# Structure–Property Correlation and Constitutive Description of Structural Steels during Hot Working and Strain Rate Deformation

**DOI:** 10.3390/ma13030556

**Published:** 2020-01-24

**Authors:** B. Gangadhara Prusty, Amborish Banerjee

**Affiliations:** 1ARC Training Centre for Automated Manufacture of Advanced Composites (AMAC), School of Mechanical and Manufacturing Engineering, UNSW, Sydney, NSW 2052, Australia; g.prusty@unsw.edu.au; 2School of Mechanical and Manufacturing Engineering, UNSW, Sydney, NSW 2052, Australia

**Keywords:** plain carbon steel, structural steels, dual-phase (DP) steel, strain rate, hot working, shear bands, constitutive equations, dynamic recrystallization (DRX)

## Abstract

The behaviour of plain carbon as well as structural steels is qualitatively different at different regimes of strain rates and temperature when they are subjected to hot-working and impact-loading conditions. Ambient temperature and carbon content are the leading factors governing the deformation behaviour and substructural evolution of these steels. This review aims at investigating the mechanical behaviour of structural (or constructional) steels during their strain rate (ranging from very low to very high) as well as hot-working conditions and subsequently establishing the structure–property correlation. Rate-dependent constitutive equations play a significant role in predicting the material response, particularly where the experiments are difficult to perform. In this article, an extensive review is carried out on the merits and limitations of constitutive models which are commonly used to model the deformation behaviour of plain carbon steels.

## 1. Introduction

It has been fundamentally established that materials deform in a different manner when subjected static and dynamic loading [1,2,3,4,5,6,7]. In many industrial applications, the dynamic response of a material is an important area of research. When a material is subjected to external forces, the response of the material depends on the (a) type of loading, (b) nature of loading, (c) effect of temperature, and (d) rate of loading. Strain rate or rate of loading is defined as the rate of change of strain with respect to time. A material behaves in a different manner with the change in the regimes of strain rates. In general, strain rates are categorized into five regimes: creep, quasi-static, intermediate, high, and ultrahigh [8,9], as depicted in Figure 1.

Engineering materials experience a wide range of strain rates and high temperatures during metalworking operations. To withstand extreme conditions such as creep deformation during rotation of turbine blades to man-made hazards like blast and impact, various techniques have been developed for enhancing the performance and reliability of mechanical structures [10,11,12]. During all these events, the effect of strain rate plays a significant role in determining and analysing the deformation behaviour of the materials. At the same time, the engineering components must also be designed to perform well over different regimes of strain rate and temperature for structural integrity. Previously, understanding the flow behaviour of the material at different strain rates was constrained only to the military, nuclear, and ballistic structures/fields, where safe design and performance of the material is of paramount importance. However, with the advancement of experimental techniques and computational tools, research towards improving the mechanical properties of the materials and predicting the dynamic plastic flow of the materials is becoming an important aspect of consideration. Various mechanical deformation processes and the corresponding strain rates associated with these processes are depicted in Figure 2. 

The figure presents a graphical depiction of how the strain-rate testing of engineering materials is important for metalworking operations and engineering structural applications and for improving the manufacturability of the engineering material [13,14]. With the recent development in the field of strain-rate testing, the flow behaviour of the materials can be accurately predicted and correlated to their actual working conditions. The test methods which are widely used for performing the experiments at different regimes of strain rates are presented in Table 1 [14].

The deformation behaviour of any material is primarily influenced by its loading conditions, especially when it is loaded dynamically. Dynamic loading consists of impacting, crushing, spalling, shear localization high-speed machining, energy absorbing of the materials, and explosive forming [15]. Furthermore, with the increase in the intensity of dynamic loading or velocity, the potential for damage and subsequent reduction in mechanical performance is increased [16]. Flow stress is an important parameter to be considered when a material is subjected to dynamic loading as the material to undergo plastic deformation. The flow stress of a material is influenced by several factors such as (a) lattice resistance, (b) presence of different phases, (c) chemical composition of the material, (d) strength of dislocation, (e) hardening due to second phases, (f) size of grain boundaries, and (g) interstitial and solute hardening [15,16,17,18,19]. The flow stress comprises the frictional term (*σ*_o_), an athermal term (*σ*_a_), and the thermal term (*σ*_p_). The frictional term (*σ*_o_) stands for the stress required to overcome the lattice friction and is dependent on strain rate and temperature. The athermal term (*σ*_a_) is considered the internal stress that occurs in the material due to the long-range barriers to dislocation motion [18,20]. The thermal term (*σ*_p_) is more sensitive to temperature and strain rate [21]. Mathematically it is expressed as follows: (1)$$\sigma =\sigma_{a}+\sigma_{o}\left(\dot{\varepsilon },T\right)+\sigma_{p}\left(\dot{\varepsilon ,\;\varepsilon },T\right)$$

Along with the effects of strain rate, thermal softening of materials is also equally important to predict the deformation behaviour of the materials. In most cases, it has been observed that the flow stress decreases with increase in the temperature and depends on the rate of strain being applied [22]. The flow stress as a function of temperature, strain, and strain rate can be expressed as follows:(2)$$\sigma =\frac{2}{\sqrt{3\left(1-m\right)}}K\varepsilon^{n}{\dot{\varepsilon }}^{m}\exp\left(-\beta T\right)$$
where *n* is the strain-hardening exponent, *m* is the strain-rate sensitivity, *T* is the temperature, *ε* is the strain, and $\dot{\varepsilon }$ is the strain rate. The strain-rate sensitivity (*m*) is an important mechanical parameter as it describes the dynamic mechanical behaviour of the material [23,24]. Strain-rate sensitivity is always related to the plastic flow and is independent of the engineering stress–strain of the material. A positive strain-rate sensitivity curve indicates an increase of flow stress with an increase in strain rate whereas a negative strain-rate sensitivity indicates a decrease in the flow stress with an increase in strain rate. The strain rate sensitivity is expressed as follows:(3)$$m=\frac{ln\sigma }{ln\varepsilon }$$

The effects of strain rate and temperature are equally important during thermomechanical processing and hot working (metal formation) of metals [25,26,27,28,29]. Microstructure evolution and flow behaviour of any material play a significant role in the flow stress of the material. The behaviour of the flow stress can be correlated to hot-working processes like dynamic recovery (DRV), dynamic recrystallization (DRX), and static recrystallization (SRX) [30]. The processes of dynamic recovery and recrystallization are considered the softening mechanisms during metal deformation [31]. DRX is a significant factor for controlling the mechanical properties and microstructure evolution during hot-working phenomenon [32,33,34]. DRX occurs after the formation of dislocations attains a critical value [35,36,37]. The microstructure obtained after the hot-working process has an important effect on the final microstructure and mechanical properties of the steel [38]. Thus, it is equally important to understand the deformation mechanism of the materials under such conditions. Understanding the deformation behaviour of a material in terms of variation in strength and ductility for an applied strain rate is a significant factor for the design of structures. The mechanical properties such as yield strength, ultimate tensile strength, flow curve pattern, total elongation, etc. are greatly dependent on the variation of strain rate and temperature. Metallic materials, composites, and polymers are sensitive to strain rate, and their response varies with change in the magnitude of strain rate. Some of the alloys and metals which are much often subjected to varying strain rates are steel, aluminium alloys, nickel-based superalloys, and high entropy alloys.

Steel is considered to be the most common material which is subjected to dynamic testing [10,39,40,41,42,43,44,45]. Steel with carbon as the main alloying element and containing only residual amounts of other elements is called plain carbon steel. In general, there are many ways of classifying steel based on structures, alloy element, carbon content, etc. Different researchers have reported different ways for the classification of the steel. Although efforts have been made by (ASTM, ASM, and SAE internationals for classification of steel, there still exists a discrepancy. ASM international considers dual-phase (DP) steel and micro-alloyed steel as subcategories of High-Strength-Low-Alloy (HSLA) steels [46] whereas few researchers refer to DP steel as 1st generation advanced high-strength sheet steels (AHSS) [47]. To account this, for this review article, we have classified steel as alloy and non-alloy, which are widely termed as structural steel. Based on the carbon content, structural steel has been categorized into four groups: (a) low and mild carbon steel which has carbon content up to 0.25%, (b) medium carbon steel having carbon content from 0.25–0.55%, (c) high carbon steel having carbon content from 0.55–1.00%, and (d) ultrahigh carbon steel having 1.0–2.0% carbon content [48]. A critical review of research is also carried out on the hot-working as well as strain-rate deformation behaviour of DP steel and micro-alloyed steel. A flow chart outlining the classification of steel is depicted in Figure 3.

The addition of carbon leads to an increase in the strength and hardness of the material but, at the same time, decreases the ductility of the material. The applicability of carbon steels depends on the percentage of carbon added in the alloy. Low carbon steels are widely used in tin plates, wire products, automobile body parts, railings, chain links, etc. Medium carbon steel is the most commonly used carbon steel preferred to be used for manufacturing of shafts, couplings, gears, and axles. High carbon steels due to its excessive hardness and strength are mostly used where high abrasion resistance is essential [49] such as for the manufacture of cutting tools, punches, knives, springs, grinding balls, and sturdy wires. Ultrahigh carbon steels can be laminated with other metal-based materials to achieve superplasticity, high impact resistance, exceptionally high tensile ductility, and improved fatigue behaviour [50]. Such steels are mostly used for the nonindustrial purposes such as axles and punches. Dual-phase (DP) steels are widely used in automotive sectors due to their high tensile strength, continuous yield point, and formability as compared to other grades of steel [51,52,53,54,55]. Micro-alloyed steel consists of micro-alloying elements which act as strengthening medium and are mostly used for the manufacturing of crankshafts for automobile sectors [56,57]. The common application areas of plain carbon steel are also depicted in Figure 4.

Ferrous materials exhibit a crystalline structure, which forms during the solidification of the molten state of the material. The formation of the grains during this process contributes to the mechanical properties of the material, and any alterations in these properties are a direct consequence of the microstructural evolution. Moreover, during the ongoing deformation process of the material during different regimes of temperatures as well as temperatures, substructural evolution in the microstructure occurs, which requires an in-depth understanding to understand the micromechanisms leading to failure of the material. Thus, structure–property correlation of materials under different regimes of strain rates is important to correlate the flow stress with the substructures evolve during plastic deformation.

### Objectives and Organization

The key objectives of this review article are as follows:to summarise the state of information on the progress made in understanding the deformation behaviour of plain carbon steels, DP steels, and micro-alloyed steels at different strain rates and during their hot-working conditions along with the experimental methodology andto discuss the current research challenges and future research which require attention in this field with a special emphasis on the research gap in the strain-rate behaviour of high carbon steels.

Review on the strain-rate effects and its subsequent microstructure evolution in low, medium, and high carbon steel along with dual-phase and micro-alloyed steels are presented in Section 2, whereas Section 3 deals with the constitutive modelling work carried out to date on plain carbon steel, DP steel, and micro-alloyed steels. The current challenges and the scope for future research are projected in Section 4. Effects of strain rate and temperature on AHSS steels such as Twinning-Induced Plasticity (TWIP) and Transformation-Induced Plasticity (TRIP) steels are not reviewed in this article. It is also important to mention here that, in case of DP steels, it is fully understood that the deformation mechanisms in ferrite-based DP steels are much different than austenite-based DP steels. Therefore, adequate care has been taken to describe the deformation mechanisms in DP steels. Based on the extensive review of articles, it may be highlighted that the earlier studies are limited to understanding of the strain-rate behaviour of the dual-phase, low, and medium carbon steels only. Research work related to the strain-rate response of high carbon and ultrahigh carbon steel in the open literature is very scarce, which could be potentially due to the lack of interest in the field of intermediate and high strain rate. As a result, the generic discussion varied in all the three different sections of low, medium, and high carbon steels. 

## 2. Structural Property Correlation

### 2.1. Low Carbon Steel

Low carbon steel, often termed as mild steel, is the most widely used steel among all the available grades. The pioneering works done on the strain-rate behaviour of low carbon steel are discussed in this section. The true stress–strain behaviour of low carbon ferrite-cementite (FC) steels at different strain rates varying between 3.3 × 10^−1^ and 5.0 × 10^−4^ s^−1^ and with different ferritic grain sizes from 0.5 to 34 μm was studied by Tsuchida et al. [58]. They showed that, with an increase in the strain rate, the stress (σ), strain (ε), and work-hardening rate were found to be increased for each of the FC steels. The authors further concluded that grain refinement up to 0.8 μm increased the tensile properties and the σ-ε behaviour of the low-carbon FC steels. Figure 5 shows the variation of the σ-ε behaviour with the change in the strain rate.

The scanning electron microscope (SEM) images of the FC steel with different grain sizes of ferrite (shown in Figure 6) demonstrated an elongation in the ferrite grains along the tensile direction, and this elongation degree was found to be the same for both the strain rates. 

Similar experimental studies were performed by Paul et al. [22] to predict the dynamic flow behaviour of low carbon and ultralow carbon steel at different regimes of strain rates from 0.0007–250 s^−1^. The authors reported an increase in the yield strength with an increase in the strain rate. However, their studies report that the strain hardening rate was observed to be decreased with an increase in the strain rate, as shown in Figure 7. The strain hardening rate depends on various factors such as the interaction between the dislocations as well as the dislocation density of the material. An overall decrease in the strain-hardening rate thus indicates the dominant role of dislocation annihilation as compared to their multiplication and interaction. The compressive flow behaviour of AISI 1018 low carbon steel at three different strain rates (10^−3^, 1, and 3.5 × 10^3^/s) was investigated by Korkolis et al. [59]. The authors, in addition to the higher flow stress with the increasing strain rates, also observed deformation-induced thermal softening mechanism at higher strains. Zheng et al. [60] studied the tensile behaviour of HRB500E low carbon steel (0.249% C) at low, medium, and high strain rates and have reported the rate-sensitivity behaviour of the material. The authors further reported a linear relationship between the dynamic increase factor for yield strength and strain rate. Lin et al. [61] conducted tensile tests in the medium strain-rate regime (2 to 75/s) for four different grades of low carbon steels (HPB235, HRB335, HRB400, and HRB500). The authors observed an increase in the yield strength from 13% to 41% as well as an increase in the ultimate strength from 9% to 19% within the tested strain-rate regime. 

Campbell and Ferguson [62] conducted high strain-rate experiments in the double shear mode to correlate the temperature and rate sensitivity of the mild steel for different ranges of temperature (196 to 713 K) and strain rates (10^3^ to 4 × 10^4^ s^−l^) and reported an increase in the flow stress with an increase in the strain rate. The authors also reported that, due to the viscous resistance to the motion of dislocation, the rate sensitivity of the steel was found to be varied as a decreasing function of temperature. Similar findings on the strain rate and flow stress relationship were also reported by Klepaczko [63]. 

The presence of different phases in multiphase steel also alters its dynamic flow behaviour at different regimes of strain rates. For instance, in case of austenite-martensite dual-phase (DP) steels, the deformation-induced austenite to martensitic transformation (DIMT) at different loading conditions plays a significant role in altering the mechanical strength as well as the ductility of the material. Moreover, at elevated temperatures, due to the tempering of martensite, the formation of ferrite and carbides or cementite is also possible, which further alters the flowability and the strain-hardening behaviour of the material. For austenite-ferrite DP steels, thermomechanical control processing (TMCP) is mostly carried out in order to maximize the grain boundary area of austenite per unit volume, which further leads to an increase in the nucleation site density for austenite to ferrite transformation [64,65]. Ok and Park [66] investigated the dynamic deformation behaviour of plain low carbon steel at a strain rate of 0.01 s^−1^. The authors found three different patterns of flow curves with the change in the temperatures. At a temperature above Ae3 (825 °C), the flow curve showed a peak behaviour, which signifies the fact that dynamic recrystallization of austenite occurred whereas, below the Ae3 temperature, the flow curve exhibited a saturated profile rather than any peaks. With the further decrease in temperature below T_0_ (780 °C), the flow curve exhibited an increase in the yield strength of the material. The authors further studied the structure–property correlation to understand the micromechanism leading for such changes in the flow curves. They observed that massive ferrites were formed when the deformation of low carbon steel within the 2-phase (γ + α) field took place below T_0_ whereas, above this temperature, the ferrites were found to be divided into a large number of subgrains by conventional strain-induced transformation. The growth of ferrite grains depending on the deformation temperature is shown in Figure 8. The ferrite in the SEM images appears to be dark. The dynamic transformation of γ → α is also referred to as deformation-induced ferrite transformation (DIFT) and is in general considered as a solid-state transformation. This process results in the formation of ultrafine ferrite grains and thus increases the strength of the material as per the Hall–Petch equation. Similar results were also reported by Chung et al. [67] for different regimes of strain rates. Rizhi et al. [68] observed two distinct processes of microstructural evolution of ferrite during compression of ferrite-austenite low carbon steel. Formation of low-angle grain boundaries was observed at low strain rates (0.0002/s) whereas, at high strain rates (0.2/s), banded substructures were formed. The authors further reported that, at high strain rates, the band structures were transformed to equiaxed grains.

The grain size is considered of significant interest for predicting the deformation behaviour of the steels at different regimes of strain rate. It is fully understood that smaller grain size leads to an increase in the grain boundaries in the metal matrix. These grain boundaries, in turn, provide a restriction to the dislocation movement during plastic deformation and thus lead to an increase in the strength of the material [69,70,71,72]. The Zener Holloman parameter [73,74,75] is mostly used to predict the resulting grain size (*Z* = $\dot{\varepsilon }$exp(*Q*/*RT*)), and the size of the recrystallized ferrite (*d*) is mathematically expressed in terms of the *Z* parameter as follows:(4)$$d=AZ^{-0.16}$$
where $\dot{\varepsilon }$ is the strain rate, *Q* is the deformation activation energy, *T* is the deformation temperature, *R* is the universal gas constant, and *A* is a constant. According to this equation, it is expected that the higher *Z* values would lead to the finer grains and vice versa [76]. Many researchers have shown the variation of the grain size with the *Z* parameter at different strain rates [77,78,79]. Murty et al. [77] studied the deformation behaviour of coarse grain ultralow carbon steel by performing experiments at nominal strain rates of 1 and 0.01 s^−1^ and reported that the ferrite grain size (*d*) and the *Z* parameter satisfy Equation (4) with a constant value of *A* being 300. Based on this correlation, the authors established the fact that diffusion along the grain boundaries was the major rate-controlling mechanism for the ferrite grain growth in such materials when processed through large strain and high Z deformation. Figure 9A shows the boundary maps, whereas Figure 9B shows the crystallographic orientation distribution of the deformed specimen for a nominal strain rate of 1 s^−1^ at different regimes of temperature.

The orientation maps of the specimens deformed at different strain rates and the variation of grain size with the Z parameter are shown in Figure 10.

Murthy et al. [77] performed similar studies and conducted plain strain-compression tests at different strain rates of 0.01 as well as 1 s^−1^ and temperature range of 773–923 K to understand the formation of ultrafine grains. They constructed a processing map and attributed grain subdivision with dynamic recovery for high Z parameter and grain subdivision with dynamic recrystallization at the low value of the Z to be the major reason for ultrafine grain formation. The thickness of the ferrite grains at a given strain rate was predicted by the following:(5)$$TH_{\alpha }=d_{\alpha }\exp\left(-\varepsilon \right)$$
where $TH_{\alpha }$ is the thickness of the ferrite grain after deformation, $d_{\alpha }$ is the initial ferrite grain size, and ε is the compressive strain applied. In another study on ultralow carbon steel [78], the authors confirmed the occurrence of dynamic recrystallization in ferritic iron deformed at different strain rates with the help of Transmission Electron Microscopy (TEM) studies. Figure 11 shows the transmission electron microscope (TEM) images of the specimen deformed at 1 and 0.01/s strain rates where CA denotes the compressive axis.

Rajput et al. [80] studied the hot-deformation behaviour of AISI 1010 steel at different regimes of strain rates (0.01–20 s^−1^) and temperature ranging from 750–1050 °C. They correlate the variation in flow stress with the change in microstructure and Zener–Hollmann parameter and reported instability in the flow stress at higher strain rates as shown in Figure 12. The true stress of the flow curve for a constant strain rate was found to be gradually decreased with an increase in the temperature.

Earlier studies on the hot-deformation behaviour of low carbon steel have been carried out at wide ranges of temperatures and strain rates [81,82,83,84,85]. Rao et al. [86] conducted compression tests on 0.06% C steel at different strain rates of 0.1, 1, and 8/s and reported a hyperbolic-sine relationship for representing the deformation behaviour in the austenitic field. Eghbali et al. [87] performed torsion tests on plain carbon steel to study the strai- rate effect on the grain refinement of ferrite and reported the development of fine new ferrite grains with high angle boundaries (HAB) with an increase in strain rate, thus showing an increase in their peak stress and steady-state stress. Furthermore, they observed the annihilation of dislocations at a low strain rate due to the diffusion effect. The effect of prestrain, temperature, and strain rates on the dislocation cell formation on low carbon steel sheet was studied by Johnson et al. [81]. The specimens were prestrained at two different conditions at different strain rates of 10^−4^, 10^−3^, and 5 × 10^−2^ s^−1^, and no change in the work hardening behaviour was observed in spite of the presence of dislocation cell structure. This finding is contrary to the general concept where the strain-hardening rate of the material decreases significantly with the formation of dislocation cell. During the process of plastic deformation, interaction and tangling of dislocation are observed in metallic materials, which in turn results in the formation of dislocation cell. The cell structures of these dislocations are furnished with high dislocation density. The formation of dislocation cells during different strain rates and its effect on the flow behaviour of steel have been reported by many researchers [88,89,90,91,92,93,94].

Rajput et al. [82] conducted hot compression tests on AISI 1060 steel in Gleeble 3800 simulator at strain rates from 0.01–80/s as well as at different temperatures under the vacuum of 1 pascal. The authors concluded the following: (a) the high values of stress exponent (*n*) and strain-rate sensitivity (*m*) were due to the dynamic recovery and recrystallization of ferrite and austenite respectively, (b) the combined effect of adiabatic heating and inhibited restoration leads to damage at the austenite triple grain boundary at high strain rates, and (c) the deformation of the steel was found to be a diffusion-controlled process as the value of apparent activation energy (290 KJ/mole) was very close to the bulk self-diffusion energy of the austenite (270 KJ/mole). The variation of the flow stress versus temperature plots and flow stress versus strain rate for all regimes of strain rates and temperatures is shown in Figure 13 [82].

Gao et al. [95] conducted a series of compression tests on a bimetal consisting of pearlitic and low carbon steel on Gleeble 3500 mechanical simulator from temperature ranges from 800–1100 °C and strain rates of 0.02, 0.1, 1, and 10 s^−1^. The authors correlated the profile of the strain-hardening region of the flow curves with the dislocation multiplication as well as interaction and the softening region of the flow curve with the dynamic recrystallization. Zhi-Xiong et al. [96] carried interrupted hot tensile tests at different strain rates from 1 to 3000 s^−1^ as well as temperature from 800–1200 °C and reported an increase in the volume fraction of the recrystallized grains with an increase in the temperature and strain rate. Similar types of isothermal compression tests were done by Wang et al. [97] at different strain rates ranging from 0.01 to 0.5 s^−1^ in Gleeble 3500 simulator at temperature of 0.01–0.5 s^−1^ to understand the deformation behaviour of carbon structural steel (Q235A) having the chemical composition of 0.17% C, 0.22% Si, 0.68% Mn, 0.0095% P, 0.006% S, and rest FE. They concluded that a decrease in the strain rate along with an increase in the deformation temperature inhibits the occurrence of dynamic recrystallization (DRX). 

Stress-relaxation phenomenon is observed when the tests are interrupted without unloading the specimen at different strain rates. These tests are widely used for characterising parameters such as internal stresses and activation volume during mechanical deformation [98]. During stress relaxation, the elastic strain gets converted to plastic strain. Stress relaxation tests were conducted at predefined engineering strains for low carbon steel and the other two grades of steel [99]. They concluded the stress drop to be inversely proportional to the rate of change of dislocation velocity. The stress drop was observed to be decreased with strain as shown in Figure 14. Strain hardening is considered the predominant medium for resistance to dislocation for low carbon steel with a ferritic phase. Thus, with an increase in the strain accumulation in the material, there is an increase in strain hardening and a subsequent decrease in stress drop.

Earlier studies done by Tsuchida et al. [100] on ferrite-cementite low carbon steel showed an increase in the lower yield point and flow stress with a decrease in the grain size at different strain rates of 3.3 × 10^−4^ s^−1^, 100 s^−1^, and 10^3^ s^−1^, as shown in Figure 15. They reported an elongation in the Lüders band at the early stages of deformation and attributed this as a reason for the decrease in flow stress with the increase in grain size. Sun et al. [101] developed and verified an explicit equation for annealed mild steel correlating the strain rate and the Lüders strain by measuring the propagation of a Lüders band, expressed as follows:(6)$$\dot{S_{L}}=\frac{el_{o}}{N\varepsilon_{L}}$$
where $\dot{S_{L}}$ is the Lüders band velocity, *ε_L_* is the Lüders strain, and *N* is the number of bands. It is to be mentioned here that $\dot{S_{L}}$ and *ε_L_* are strongly rate dependent, and thus, the Lüders band velocity is a strain-rate-dependent property.

Development of localized shear bands and the concurrent microstructure evolution in a 0.22% C steel at different strain rates of 610, 650, and 1500 s^−1^ was investigated by Xu [102] while performing torsional testing using split Hopkinson bar. Three different heat treatments were given to the steel viz (a) heating at 850 °C and holding for 30 min followed by air-cooled; (b) heating at 850 °C, holding for 30 min, and then quenched in water; and (c) heating and holding at 850 °C for 30 min, followed by quenching, and then tempering at 300 °C for again 30 min. They concluded (a) that, the higher the strength of the steels, the easier is the formation of the shear bands; (b) that the shear localization was found to occur after the material reached a critical strain. Before arriving at the critical strain, the deformation was uniform for the entire gage length, whereas after reaching the critical strain, the deformation was localized and the material had undergone work softening; (c) that the fracture surface of all the three steel samples showed a transgranular mode of fracture, indicating a ductile failure, as shown in Figure 16; and (d) that the formation of shear localization was due to the change in the crystal orientation and initiation and growth of the microcracks.

### 2.2. Medium Carbon Steel

Steels with carbon content in the range of 0.25–55% are generally termed medium carbon steels. In addition to micro-alloyed steels, medium carbon steels are also used for structural applications, and thus, understanding the dynamic flow behaviour of medium carbon steels is important from a functional aspect. Therefore, it is significant to understand the dynamic flow behaviour of these materials, which has been discussed in detail in this section. Saadatkia et al. [103] investigated the high-temperature deformation behaviour for 0.5% carbon steels under varying strain rates (10^−4^–10^−1^ s^−1^) and temperatures from 900–1100 °C. They observed three types of flow behaviours: (a) single peak, (b) multiple transient steady-state peaks (MTSS), and (c) cyclic behaviours, as shown in Figure 17. The curves depict that, at low temperature (900 °C) and at high strain rates, the flow curves exhibited single peak behaviour whereas, at high temperature (1100 °C) and low strain rates, a multi-peak pattern was observed. 

Evolution of subsequent microstructure with various regimes of strain rates and temperature in plain medium carbon steel has been studied by many researchers [104,105,106,107]. Deformation behaviour of the plain 0.46% C steel at compression mode was studied by Li et al. [104] at five different strain rates (0.001, 0.01, 0.1, 1, and 10 s^−1^) and at different temperatures of 550, 600, 650, and 700 °C. The authors found an additional softening in the flow stress after the peak stress at higher strain rates and lower temperatures. The authors also noticed the occurrence of dynamic recovery of ferrite as well as fracture of cementite at high strain-rate deformation. On the other hand, at lower strain rates, decomposition of cementite along with ferrite recovery was observed. 

Chai et al. [105] performed compression tests on a plain 0.45% carbon steel at strain rates of 0.01, 1, and 10 s^−1^ in the temperature range of 1053–1253 K at an interval of 50 K and concluded finer DRX grain size with the increase in the strain rate. Isothermal compression tests on Gleeble 3500 were performed by Zhao et al. [106] at different strain rates ranging from 0.001–0.1 s^−1^ and at a temperature range of 550–700 °C to understand the microstructural evolution of quenched and annealed 0.45% carbon steel. The initial microstructures of the annealed samples were ferrite and pearlite, whereas for the quenched sample, it was martensite. They observed that, for the quenched samples, the flow stresses were higher than annealed ones when the strain rates were greater than 0.001 s^−1^ at 550 °C. For the rest of the conditions, the flow stress of the quenched specimens was initially higher than that of the annealed samples and then decreased after a critical value of strain. 

Duan and Zhang [107] studied the microstructural features and formation mechanisms of adiabatic shear bands in AISI 1045 steel induced by high-speed machining and observed the formation deformed shear bands (low strain rates) as well as transformed shear bands (high strain rates). The authors concluded that the deformation bands were formed due to the severe plastic shear, whereas the transformed bands were formed due to the process of recrystallization, reorientation, and elongation of the martensitic laths along with the formation of subgrains and equiaxed grains. The formation of both transformed as well as deformed shear bands were observed as shown in Figure 18. It was further concluded that the martensitic laths were elongated along the direction in the deformation bands and experienced plastic deformation only. 

Twinning formation in ferrous material is a significant feature of plastic deformation. The formation of twin planes results in a change in the orientation of the available planes, which leads to an easy slip of the material. The formation of twinning during deformation of metallic materials at high strain rate has been reported by several researchers [108,109,110]. Orawa [111] showed that the presence of carbon content had a significant impact on the formation of deformation twinning. They reported that, for material with more than 0.2% carbon, the formation of twining at strain rates between 10^2^ s^−1^ and 10^4^ s^−1^ was very scarce. Haque et al. [112] reported no noticeable difference in the microstructure of a 0.41% carbon steel, which was deformed at different high strain-rate regimes (10^3^–10^5^ s^−1^) at −30 °C, at room temperature (RT), and at 235 °C. They further concluded that the initiation of twin bands during impact was a result of the presence of the alloying elements and inclusions despite its high strain-rate sensitivity. Maiden et al. [113] developed dynamic yield criteria for medium carbon steel by conducting compressive impact tests and found it to be in good agreement with the experimental results. They reported that, at high strain rates and low-temperature regimes, the micromechanism for the deformation was the formation of fine slip lines in the ferrite region.

The effect of grain size and strain rate on the strength and strain-rate sensitivity parameter (m) of a nanocrystalline and ultrafine-grained 0.55% carbon steel was predicted by Baracaldo et al. [114] using nanoindentation techniques at various strain rates from 3 × 10^−3^ to 10^−1^ s^−1^. The strain-rate sensitivity was determined using the following equation:(7)$$m=\left[\frac{\partial logH}{\partial log\dot{\varepsilon }}\right]$$
where *H* is the hardness of the material (GPa) and $\dot{\varepsilon }$ is the strain rate. They reported a constant decrease in the *m* value for the ultrafine regime, whereas for the nanocrystalline regime, a minor increase in the *m* value with a decrease in the grain sizes was observed. Also, the strength of the material was found to be slightly increased with the increase in the strain rate, as shown in Figure 19.

Fu and Yu [115] conducted a hot compression test on a 0.36% plain carbon steel in Gleeble. They performed single hit compression tests from 0.01 to 10 s^−1^ at different temperatures and reported an increase in the flow stress with the increase in the strain rate but a decrease with the increase in the temperature except at 850 °C, as shown in Figure 20. They observed the occurrence of dynamic recrystallization after a critical strain is achieved by the material and reported a drop in the stress after attaining the peak strain *ε_p_*.

### 2.3. High Carbon Steel

High carbon steels are mostly used for industrial application in extreme operating conditions due to their hardness, strength, and relatively low cost compared to high steel alloys [49,116]. Such steels are mostly used where high abrasion is a necessity. They are mostly used for manufacturing of drill bits, masonry nails, metal cutting tools, grinding balls, and knives. In contrast to the low and medium carbon steels, after an exhaustive literature review, the author predicts that the studies on the strain-rate behaviour of high carbon steel are very scarce. This may be due to the relatively brittle nature of these steels and the lack of interest for their deformation behaviour at different regimes of strain rates and temperatures. In this section, a critical review of research work performed on the strain-rate behaviour of high and ultrahigh carbon steel is presented. 

Earlier studies have been done to investigate the effect of change in the carbon content with respect to DRX, dislocation annihilation, grain recovery, and activation energy [117,118,119,120,121,122,123,124,125,126]. Serajzadeh and Taheri [117] investigated the effect of carbon content on the DRX, flow stress, and recovery phenomenon of carbon steels during their hot deformation and reported a faster occurrence of DRX in high carbon steel as compared to the low carbon steel as shown in Figure 21a. They further concluded that the presence of carbon increases the dynamic recovery rate at low strain rates due to its effect on the process of dislocation climb and self-diffusion rate. At higher strain rates, it decreases the rate of dynamic recovery, which is presented in Figure 21b. 

Jaipal et al. [118] studied the behaviour of flow stress and dynamic recrystallization in high carbon steel at various strain rates and reported a decrease in the flow stress with an increase in the carbon content at low strain rates and high temperatures. Furthermore, at high strain rates and low temperatures, the peak flow stress of the high carbon steel was found to be more than that of the low carbon steel. Dixon et al. [119] investigated the effect of carbon and austenite grain size on the flow stress of plain carbon steel. Their results revealed that the low carbon steel possesses higher strength below strain rates of 1 s^−1^. 

Wray, in his previous studies [120,121], conducted hot tensile tests to determine the flow stress behaviour of plain carbon steels at different strain rates varying from 6 × 10^−6^ to 2 × 10^−2^ s^−1^ as a function of carbon content in the range of 0.005 to 1.54%. His studies revealed that an increase in the carbon content leads to a decrease in the work-hardening region and the flow stress of the material. The hot strength of the austenitic steels with varying percentage of carbon (0.0037 to 0.79%) was modelled by Kong et al. [122] using the artificial neural network (ANN). They confirmed the decrease in the flow stress with the increase in the carbon content at low strain rates and high temperatures, whereas the flow stress was found to be lower at high strain rates and low temperature. Lee and Liu [123] performed compressive type SHPB tests on low carbon S15C (0.15% C), medium carbon S50 (0.48% C), and high carbon SKS93 (1.16% C) steel at different strain rates of 1.1, 2.0, 2.8, 3.7 × 10^3^, and 5.5 × 10^3^ s^−1^ and temperatures of 25, 200, 400, 600, and 800 °C. The authors found an overall increase in the flow stress with an increase in the strain rate, as shown in Figure 22. As evident from Figure 22, the flow stress for SK50 was found to be 1.3 times higher than the flow stress of S15 steel. Similarly, SKS93 exhibited an approximate 10% increase in the flow stresses when compared to S50 steel. 

The temperature rise due to adiabatic heating as calculated by Lee and Liu [123] as a function of strain at different strain rates at 25 °C is presented in Figure 23. The material exhibited a rise in the temperature with an increase in the true strain values. For instance, in the case of SKS93 high carbon steel, the rise in the temperature when deformed at the strain rates of 1.1 × 10^3^ and 5.5 × 10^3^ s^−1^ were observed around ~75–80 °C. In terms of microstructural evolution, an increase in the dislocation annihilation at elevated temperatures was documented.

The superplastic behaviour of ultrahigh carbon steel different strain rates for three different percentages of carbon (1.3, 1.6 and 1.9%) at different temperature regimes was investigated by Sherby et al. [124]. For both the strain rate change tests as well as stress relaxation tests, the measured values of strain-rate sensitivity exponents at 650 °C were reported to be 0.35–0.40 for 1.3% C steel, 0.40–0.45 for 1.6% C steel, and 0.40–0.50 for 1.9% C steel. The tests performed at 750–850 °C revealed the value of strain-rate sensitivity for all the three steels to be around 0.40–0.45. 

It is well understood that, during impact and high strain-rate loading conditions, the material is incapable of releasing the heat generated during the process of deformation as that in the case of quasi-static tests [127,128,129,130,131,132]. Thus, the ongoing deformation process is considered adiabatic rather than isothermal in nature. The heating during the adiabatic process may significantly affect the flow behaviour of the material and needs proper investigation. Since the deformation of material exhibits an adiabatic process at high strain rates, a substantial amount of the plastic work gets converted into heat and, thus, the material attains an increase in temperature. The rise in temperature in ferrous material is calculated by the following equation:(8)$$\Delta T=\frac{\eta }{\rho C}\int \sigma d\varepsilon$$
where *ρ* is the density of the steel (7850 kg/cm^3^); *C* is the specific heat (0.49 kJ/kg/°C); and *η* is the proportion of the plastic work which is converted into heat, which was taken to be 100%. In another study [126], Lee and Liu investigated the adiabatic shearing behaviour of the same steels viz S15C low carbon steel, S50C medium carbon steel, and SKS93 high carbon steel at two different strain rates of 5 × 10^4^ and 2 × 10^5^ s^−1^ using hat-shaped specimens in SHPB testing technique. The authors reported that the shear flow stress, width, and hardness of the shear band were strongly dependent on the amount of carbon content and the strain rate. They observed the formation of deformed and martensitic shear bands in medium carbon and high carbon steel. However, for low carbon steel, only the deformed shear bands were observed. Dimples were observed on the fracture surface of low carbon steel only, whereas for both the medium and high carbon steel, the fracture surface exhibited both dimples and knobby features. The fracture surfaces of all the three deformed steel samples at different strain rates are presented in Figure 24. 

A similar study was done by Nakkalil [133] to investigate the formation of adiabatic shear bands in plain 0.77% carbon steel during high strain-rate compression testing at different temperatures. The authors concluded that the formation of adiabatic shear band occurs due to the strain localization, which is a common phenomenon during discontinuous load drop. They further observed that, at constant temperature, an increase in the strain rate leads to a decrease in the critical adiabatic strain. On the other hand, an increase in temperature at constant strain rate results in a decrease in the formation of adiabatic shear bands (ASBs). Likewise, in previous studies [107,124], both deformed and transformed bands were observed. Moshksar and Rad [134] analysed the superplastic behaviour of heat-treated fine-grained 0.9% plain carbon steel by conducting experiments at a strain-rate range of 5 × 10^−5^ s^−1^ × 10^−3^ s^−1^ and at a temperature range of 650–710 °C. They reported a shift in the strain-rate sensitivity exponent (*m*) towards the greater strain rates with an increase in temperature, as shown in Figure 25. It is shown that, for all the strain rates, the total strain reaches a peak point at a threshold temperature value and then it drops. They reported that, at low strain rates, the grain growth starts at a relatively lower temperature because the specimen is subjected to test temperature for a longer period and thus gets sufficient time for grain growth. With further increase in the temperature, the grain growth accelerates, which leads to the decrease in the total strain. 

Limited research in the open literature is available to discuss the strain-rate deformation behaviour of ultrahigh carbon steel (UHC) at different regimes of temperature [135,136,137]. Ozdemir and Orhan [135] investigated the superplastic deformation behaviour of 1.18% carbon steel at various strain rates as well as temperatures (650, 735, and 850 °C) and reported that the material achieved the best superplasticity within the strain rates between 3.3 × 10^−3^ and 4.6 × 10^−3^ s^−1^ and at 735 °C. Zhang et al. [136,137], in their studies, performed compression tests of ultrahigh carbon steel in Gleeble 3500 at different strain rates to investigate the evolution of spheroidization at elevated temperatures. The authors observed that low strain rate encouraged the formation of spheroidal cementite. A similar study was done by Harrigan and Sherby [138] to investigate the effect of strain rate on the evolution of spheroidization of 0.79% carbon steel. They performed compression tests on cylindrical specimens at three different strain rates of 0.0003, 0.003, and 0.03 s^−1^ and five different sets of temperatures: 550, 550, 600, 650, and 700 °C. The degree of spheroidization was found to be dependent on both the strain rate as well as temperature in the form of $\dot{\varepsilon }e^{\frac{Q}{RT}}$. This was later confirmed through microstructural investigation. Additionally, it was further concluded that the activation of the strain-induced diffusion paths (grain boundaries) along with vacancies and dislocation movement at high strain rates leads to an enhancement in the spheroidization rate.

The deformation behaviour and the subsequent microstructural evolution in 1% carbon steel at ambient at different temperature regimes have been investigated by Banerjee et al. [139,140,141,142,143]. The authors reported an increase in the yield strength of the material with an increase in the quasi-static strain rate at all the temperatures. Additionally, the strain-hardening behaviour of the material exhibited an overall decreasing trend with an increase in the temperature, as shown in Figure 26. In their another study, the authors investigated the tension–compression asymmetric behaviour of 1% C steel at quasi-static strain rates and reported the variation in the DIMT phenomenon under tensile and compressive loading [144]. The DIMT phenomenon was reported to be rate dependent for tensile loaded specimens, whereas the phenomenon was found to be rate independent for compressive loaded specimens. The authors correlated this phenomenon with the variation in the molar volume as well as hydrostatic stresses developed during tensile and compressive loading. The high strain-rate deformation behaviour of high carbon (1%) steel at different temperatures (25, 100, and 175 °C) was also investigated by Banerjee et al. [142] using split-Hopkinson pressure bar testing machine. The authors reported an irregular trend in terms of ultimate strength and elongation with respect to temperature. 

### 2.4. Dual-Phase (DP) Steel and Micro-Alloyed Steel

Dual-phase (DP) steel refers to the class of high-strength steel consisting of dispersed bainite or martensite in the soft ferrite matrix [145,146,147]. Martensite/bainite contributes to the hardness, whereas ferrite adds to the ductility of the DP steel. This microstructure results in an excellent combination of strength and ductility. The strain-hardening rates and energy-absorption capabilities of DP steels are also higher than the conventional high strength steel (HSS) grades of steel. DP steel possesses high strain-rate sensitivity and low yield to tensile strength ratio and is mostly used in automobile sectors. These steels are considered to have better deformability than other grades of AHSS steel with similar strength [148]. The mechanical properties of DP steel have been studied by many researchers [149,150,151,152,153,154]. Bag et al. [155] reported an excellent impact toughness of DP steel when the volume fraction of martensite is around 55%. Saeidi et al. [156] showed that the elongation and Charpy impact energy for bainite-34% ferrite dual-phase steel were found to be higher than the bainite and bainite-ferrite microstructures. Modi [157] reported that the volume fraction of ferrite and martensite significantly affect the wear characteristics of DP steel. In this section, the research work done on strain-rate behaviour of DP steel is critically reviewed.

Earlier studies have been carried out to investigate the effect of temperature and strain rates on the deformation behaviour of DP steels [158,159,160,161,162,163,164,165,166]. Cao et al. [158] performed tensile tests at varying strain rates (10^−4^ to 10^2^ s^−1^) and temperature ranging from −60 to 100 °C on DP800 grade steel. The authors reported an increase in the yield as well as the ultimate tensile strength of the material with an increase in the strain rate and decrease in the temperature. Similar findings have been reported by other researchers [160,161,163]. Yu et al. [159] conducted quasi-static strain rate tests from 10^−4^ to 10^−2^ s^−1^ and dynamic tensile tests at strain rates 500, 1100, and 1600 s^−1^ for DP 600 steel and observed the yield strength at high strain rates to be approximately twice that in the quasi-static strain rates, as shown in Figure 27. 

Tarigopula et al. [162] reported an increase in the dynamic flow rate of DP800 steel (C: 0.12%, Si: 0.20%, Mn: 1.50%, P: 0.015%, S: 0.002%, Nb: 0.015%, and rest FE) steel with the change in the strain rates. The authors also observed severely localized strains at high strain rates. Sachdev [164] investigated the effect of retained austenite (RA) on the deformation behaviour of DP steel and reported that the presence of RA significantly affects the strain-hardening exponent of DP steel. Dai et al. [165] reported two stages of a strain-hardening mechanism for DP1180 steel for strain rates ranging from 10^−3^ to 1750 s^−1^. The authors further observed the formation of dislocation cell blocks of 90 nm size and adiabatic temperature rise for higher strain rates. Rahmann et al. [166] conducted shear tests at varying strain rates from 0.01 to 600 s^−1^ for DP600 steel at room temperature. The authors reported that, beyond 50% of the shear strain, the steel showed negative strain-rate sensitivity at higher strain rates of 100 and 600 s^−1^, as shown in Figure 28. 

Hassannejadasl et al. [167] reported alteration in the flow surface for DP600 steel with a variation in the material anisotropy coefficients when subjected to various strain rates from 10^−3^ to 10^3^ s^−1^. Huh et al. [168] investigated the effect of strain rates ranging from 0.003 to 200 s^−1^ on the deformation behaviour of DP600 and DP800 steel and found an increase in the flow stress with an increase in the strain rate. Misra et al. [169] conducted nano-indentation tests at different strain rates from 0.05 to 1 s^−1^ for an ultrafine Fe–0.95C–1.30Mn–0.91Si–0.23Cr DP steel at room temperature and observed a high strain-rate sensitivity with twinning as the major controlling mechanism for the deformation of material. Samuel et al. [170] described the strain-hardening behaviour of uniaxially deformed dual-phase steel by a modified Crussard-Jaoul (C-J) analysis and reported an increase in the yield strength, ultimate tensile strength, and work-hardening rate with an increase in the strain rate. 

The effect of strain rates on the fracture and deformation behaviour of DP600 base metal (BM) containing 0.061% C and its welded joint (WJ) was investigated by Dong et al. [171]. The authors carried out quasi-static and dynamic tensile tests at varying strain rates extending from 0.001 to 1133 s^−1^. The yield as well as the ultimate strength of the material exhibited an increasing trend with an increase in the strain rate, as shown in Figure 29. However, the author observed no significant change in the fracture behaviour for the base as well as the weld material.

Mocho et al. [172] investigated the deformation behaviour of DP500 steel at quasi-static and dynamic strain rates after initial fatigue loading and reported the localized strains for high strain rates to be higher than the quasi-static tests. Joo et al. [173] investigated the tension/compression-hardening behaviour of DQ (drawing quality) and DP590 steel of 1 mm thick at various strain rates from 0.001 to 50 s^−1^. The flow stress of the material was found to be increased with an increase in the strain rate both in tension and compression. It was further noticed that the strain-rate sensitivity of DP590 in compression was more sensitive than that in tension. 

Several other studies have been carried out on the deformation behaviour of DP780 steel at different strain rates [174,175,176]. Huh et al. [174] investigated the effect of strain rate on the plastic anisotropy of DP780 steel and reported a reduction in the plastic anisotropy with an increase in the strain rate. They further developed a new method to quantify the r-value (plastic strain ratio) with the help of digital image correlation (DIC). The effect of stress triaxiality and strain rate (10^−3^ to 1500/s) on the deformation behaviour of DP780 steel was investigated by Anderson et al. [175]. The material exhibited an increase in the failure strain with the increase in the stress triaxiality, as shown in Figure 30. In addition, the strain-rate sensitivity was observed to be mostly positive for all conditions with a slightly negative for uniaxial specimens up to strain rates of 10^−1^ s^−1^, as shown in Figure 31.

Kim et al. [176] conducted high strain-rate experiments on DP780 and 980 steel ranging from 10^−1^ to 500 s^−1^ and observed a significant change in the yield strength and ultimate tensile strength with the change in the strain rate. Tarigopula et al. [177] conducted static and dynamic tensile tests on DP800 steel and reported an increase in the flow stress with an increase in strain rate from 10^−3^ to 500 s^−1^. 

Das et al. [178,179] investigated the deformation behaviour of DP600 and DP800 grades of steel at varying strain rates from 10^−3^ to 800 s^−1^ and found an increase in the yield strength and ultimate tensile strength for both DP600 and DP800 steel with the increase in the strain rate. It was observed that the rate of increase in the strength of these steels was higher in higher strain rate regimes (≥100 s^−1^) as compared to lower strain rates (shown in Figure 32). 

Sato et al. [180] investigated the deformation behaviour of DP590, DP 980, and DP 1180 grades of steel for varying strain rates (10^−3^, 10^1^, and 10^2^ s^−1^) The fracture strains for all the three grades of DP steel were found to be independent of the strain rate. It was further observed that the strain rate played a significant role in the strain localization for DP590 steel, whereas for DP980 and DP1180, steel the influence of strain rate on the strain localization was found to a minimum.

The effects of microstructure on the strain-rate deformation behaviour of DP steels have been carried out by many researchers [175,178,181,182,183]. Anderson [175] reported the presence of dimples and shear lips along with the formation of transverse cracks of DP800 steel when subjected to varying strain rates.

Oliver et al. [181] studied the microstructural changes for DP600 and DP800 steel when subjected to low strain rate (0.001 s^−1^) as well as high strain rate (200 s^−1^) and reported an increase in the transformation of austenite into martensite and elongation of ferrite grains at higher strain rates. The phenomenon of austenite to martensite transformation in dual-phase steel during mechanical loading is also termed deformation-induced martensitic transformation (DIMT). The DIMT phenomenon can be stress or strain induced. The presence of martensite in the material results in an overall increase in the strength of the material at the loss of the ductility. The DIMT phenomenon in different grades of steel has been earlier reported by many researchers [184,185,186,187,188,189,190,191,192,193]. 

Gündüz et al. [182] studied the effect of strain rates (from 0.005 to 950 s^−1^) on the change of yield strength of DP600 grade steel containing 0.11% C and prestrained in the range of 2 and 4% engineering strain. The authors found the change in yield strength to be insensitive to the dislocation density. In addition, the authors observed the fracture surface showing cleavage pattern for samples aged at 100 °C and dimples for the samples aged at 200 °C for both prestraining at 2% and 4%. 

Berbenni et al. [183] investigated the influence of the microstructure of DP500 and DP600 steel containing 10 and 15% martensite on its dynamic deformation behaviour when subjected to strain rates varying from 10^−3^ to 100 s^−1^ and reported that the increase in the martensitic content increases the stress with an increase in the strain rate. Queiroz et al. [194] investigated the strain ageing behaviour of a 0.10% C dual-phase steel by conducting static tensile tests at varying strain rates from 5 × 10^−4^ to 10^−2^ s^−1^ and temperature from 25 °C to 600 °C. Serrations were observed in the stress–strain curve between 155–250 °C for 10^−3^ s^−1^ strain rate (shown in Figure 33), which in turn signifies the Portevin–Le Chatelier PLC effect. 

Kim et al. [195] conducted uniaxial tests to investigate the relationship between strain rate (10^−3^ to 100 s^−1^) and formability of two steel sheets of CQ and DP590 grades. DP590 steel exhibited an increase in the flow stress with an increase in strain rate. Furthermore, the total elongation of DP steel exhibited an increasing trend for high strain rates ranging from 0.1–100 s^−1^, whereas for quasi-static strain rates (10^−3^ to 10^−1^ s^−1^), DP steel exhibited a decreasing trend. The authors attributed the change in the dislocation structure and local strain-rate hardening at high-speed deformation as major reasons for the increase in total elongation at higher strain rates. 

Micro-alloyed steels are low alloy steels containing carbides or carbonitride-forming elements such as niobium, vanadium, titanium, etc. in a small amount for refining the grain refinement and precipitation strengthening resulting in an increase in the yield strength of the steels. Micro-alloyed steels are considered a subclass of HSLA steels. Micro-alloyed steels are widely used in automobile sectors due to their higher strength [196]. 

Few researchers have studied the hot-working deformation behaviour of micro-alloyed steels [197,198,199,200]. Equbal et al. [197] investigated the effect of temperature and strain rate on the flow stress behaviour of a 38MnVS6 micro-alloyed steel during the forging operation. The authors reported an increase in the flow stress with an increase in the strain rate (0.2, 2, and 20 s^−1^) and for constant deformation temperature. However, with increase in the temperature, a decrease in the flow stress was observed. Zhang et al. [198] investigated the effect of alloying element (niobium) on the DRX behaviour of an Nb micro-alloyed steel at different strain rates and temperatures and compared the results with medium carbon steel. The authors reported an increase in the values of peak stress (σ_p_) and peak strain (ε_p_) with increasing strain rate and a decrease in temperature for bota Nb micro-alooyed steel and plain carbon steel, but the values were found to be higher in the Nb steel. Yang et al. [199] analysed the DRX behaviour of Nb–V–Ti micro-alloyed steel for strain rates varying from 0.01–10 s^−1^ and temperatures from 950 °C to 1100 °C and reported an increase in the flow stress with an increase in the strain rate and decrease in the temperature. Badjena et al. [200] studied the dynamic recrystallisation (DRX) behaviour of Al–V–N micro-alloyed steel for two different cooling conditions (air and water-cooled) at different strain rates ranging from 10^−2^ to 5 s^−1^. It was observed that, at low strain rates (0.01 to 1 s^−1^), precipitation of VCN particles occurred whereas, at high strain rate of 5 s^−1^, coarsening of AlN particles took place. The authors attributed this phenomenon as a reason towards increase in the DRX behaviour with increase in the strain rate for both air-cooled and water-cooled specimens, as shown in Figure 34. 

A critical literature review was conducted in the previous sections to understand the mechanical behaviour and deformation micromechanisms of plain carbon steels under the influence of strain rate considering the thermal effects. The substructural evolution in the microstructure using different types of characterisation tools such as TEM, SEM, EBSD, optical microscope (OM), etc. has also been reported. The next section deals with an in-depth discussion of the various types of constitutive equations developed to describe the flow behaviour of plain carbon steels. 

## 3. Numerical Modelling

Along with understanding the engineering applications of the plain carbon steels, it is equally important to fundamentally understand the flow behaviour of these materials at different strain rates and temperatures. The yield strength, ultimate strength, total elongation, plastic flow curve, work-hardening, and necking behaviour of any material depends heavily on the applied strain rates. Currently, numerical analyses are frequently adopted to acquire and predict the flow stress behaviour of materials. Since such simulation directly determines the applicability of the materials and are related to the design procedures; therefore, suitable numerical and physical models are required which can explain the strain-rate deformation behaviour. Multiscale modelling, grain growth modelling, plasticity-based analytical modelling, and finite element (FE) simulations have been often used to analyse the strain-rate effects of metallic materials during crashworthiness of auto components, thermomechanical processes [201,202], and collision of grinding balls in ball mills [203,204]. Understanding the flow pattern of the material where dynamic deformations are unavoidable and during hot-working conditions (rolling, forging, and extrusion) is important for the appropriate structural integrity of the components. The flow pattern has a direct correlation with the concurrent microstructural evolution during strain-rate deformation and kinetics of metallurgical applications during hot working such as static, dynamic, and meta-dynamic recrystallization behaviours [59,205,206,207,208]. Constitutive equations are generally used to describe the inelastic flow of the materials and are used as input for the finite element code. Thus, the preciseness of any FE-based modelling largely depends on the accuracy of the framed or chosen constitutive equations for predicting the flow behaviour of the materials at different strain rates. Constitutive equations are often used to represent flow behaviours of the metallic materials that can be used in the finite element method (FEM) simulations to model the material’s response under quantified loading conditions [209,210]. Constitutive models correlate the flow stress with the strain, strain rate, and temperature of the material. Constitutive equations are mathematically expressed as follows:(9)$$\sigma =f\left(\varepsilon ,\;\dot{\varepsilon },\;T\right)$$

Many constitutive equations have been proposed and modified in recent years to describe the abovementioned phenomenon. Each developed constitutive model has its own benefits and shortcoming depending on their applicability, computational time, and required accuracy. Basically, the constitutive equations are divided into three main categories: (a) phenomenological constitutive model, (b) physical-based constitutive model and (c) artificial neural network (ANN). Phenomenological models are mostly applicable in real-world applications where the material is subjected to dynamic loading [201] and mostly have less number of material constants. The physical-based constitutive model depends on three components: (a) athermal component, (b) thermal components, and (c) viscous drag components [211,212]. Although the physical-based models deliver more accurate results regarding the flow behaviour of the material as compared to the phenomenological models, the physical models are less preferred because of their large material constants and their requirement of large and precise data for finding the value of material constants. Most of these models show a rise in the flow curve with an increase in strain rate and a decrease in temperature [205]. For the robustness and effectiveness during formulation/modification of a dynamic flow model, the following five parameters must be considered: (a) strain rate range, (b) strain-rate sensitivity effects, (c) material constants and parameters, (d) effect of temperature, and (e) prediction of flow at high stress. This section deals with a review of the constitutive equations which has been developed/modified and has been reported for plain carbon steels when they are subjected to various strain rates and hot-working conditions. 

Li et al. [213] predicted the flow behaviour of B1500HS low carbon steel when subjected to compression tests at different strain rates (0.005 to 10/s) as well as temperature ranging from 700–900 °C. The developed model was found in good agreement with the experimental results. Lee and Liu [123] predicted the flow stress behaviour of low (0.15% C), medium (0.48% C) and high carbon (1.16% C) steel at two different sets of strain rates and temperature regimes by applying the Zerilli–Armstrong (ZA) body-centered cubic BCC model [214,215]. The Zerilli–Armstrong equation is mathematically represented as follows:(10)$$\sigma =c_{0}+c_{1}\varepsilon_{pl}^{n}+c_{2}\exp\left[\left(-c_{3}+c_{4}ln\dot{\varepsilon }\right)T\right]$$
where *c*_0_ is the athermal stress component, *c*_1_ is the work hardening coefficient, *c*_2_ is the thermal stress coefficient, *c*_3_ is the coefficient of thermal softening, *c*_4_ is the coefficient of temperature and strain-rate coupling term, and *n* is the work hardening exponent [123]. The calculated values of all the coefficients are presented in Table 2. They reported that the error between the calculated flow stress and the experimental results was less than ±5%, as shown in Figure 35. Similar results have been reported by Lee and Liu in their other study on the dynamic compression study for medium carbon steel [216].

Paul et al. [22] observed a significant decrease in the strain-hardening rate with strain rate for micro-alloyed, low, and ultralow carbon steels. Based on their observation, they proposed a strain-rate-sensitive model for predicting the flow behaviour during the automotive crash event and analysed that the model performed well for strain rates varying from 0.0007–250 s^−1^. They expressed the constitutive equation as follows:(11)$$\sigma =\sigma_{o}\left(1+Aln\left(\frac{\dot{\varepsilon }}{{\dot{\varepsilon }}_{o}}\right)\right)+(B\varepsilon_{p}+C\left(1-exp\left(-\beta \varepsilon_{p}\right)\right)\left(1-Gln\left(\frac{\dot{\varepsilon }}{{\dot{\varepsilon }}_{o}}\right)-H\sqrt{\frac{\dot{\varepsilon }}{{\dot{\varepsilon }}_{o}}}\right)$$
where *A*, *B*, *β*, *k*, *C*, *G*, *H*, and *σ_o_* are the material constants. *σ_o_* is the yield strength which was calculated from the engineering stress–strain curve. The material constants *B*, *C*, and *β* were calculated by fitting the curve with the quasi-static stress–strain curve. The remaining material constants *A*, *K*, *G*, and *H* were determined from the yield stress vs. strain rate plot, as shown in Figure 36. They further justified their model and compared the calculated results with the experimental results and other available models to predict the variation in yield strength of low and ultralow carbon steel with a change in the strain rate. The calculated values of constants are presented in Table 3 for both ultralow and low carbon steel.

Majzoobi [217,218] conducted both tensile and compression experiments on two different plain carbon steels with carbon contents of 0.26 and 0.165, respectively, and used a combination of numerical and optimisation approaches to predict the material constants of the Johnson–Cook (JC) [219,220], Zerilli–Armstrong (ZA), and Power-Law-Plasticity (PL) [221] models. The mathematical expression for both the JC model and PL model are described below: 

Johnson-Cook model:(12)$$\sigma =\left(A+B\varepsilon^{n}\right)\left(1+Cln\left(\frac{\dot{\varepsilon }}{{\dot{\varepsilon }}_{o}}\right)\right)\left(1-{\left(\frac{T-T_{r}}{T_{m}-T_{r}}\right)}^{m}\right)$$
where *σ* is the equivalent flow stress, *A* is the yield stress at the reference strain rate and temperature, *B* is the strain hardening coefficient, *C* is the coefficient of strain rate hardening, *T_r_* is the reference temperature, *T_m_* is the melting temperature, *m* is the thermal softening exponent, and *n* is the strain hardening exponent. 

Power-law model:(13)$$\sigma =K\varepsilon^{n}\varepsilon^{˙m}$$
where *σ* is the equivalent flow stress, *K* is the strain hardening coefficient, *n* is the strain hardening exponent, and *m* is the strain rate sensitivity. Majzoobi et al. [217,218] reported that, unlike the JC model and ZA model which can accurately predict the flow behaviour of these steels, the PL model was not an appropriate model to depict the deformation behaviour. They attributed the absence of temperature effect in the PL model as a reason for this discrepancy in the result.

Buckingham π theorem has a special application in applied engineering problems. The theorem states that, for a mathematical equation having *n* number of variables, the actual expression can be expressed in terms of dimensionless parameters *π*_1_, *π*_2_, *π*_3_ … *π_n_* [222]. Based on this theorem, Phaniraj and Lahiri [223] developed a constitutive equation for predicting the deformation behaviour of different grades of plain carbon steel with carbon contents from 0.005 to 1.54% and at strain rates from 6 × 10^−6^ to 2 × 10^−6^ s^−1^. Spirdione et al. [224] investigated the flow stress behaviour for two different types of plain low carbon steel: (a) coarse-grained material having 9% pearlite and α-ferrite phases and (b) heat-treated condition where there was no pearlite by conducting high strain-rate experiments in an SHPB apparatus of the order 10^2^–10^4^ s^−1^ and at different temperatures varying from RT to 650 °C. Based on the thermal activation theory, the authors build a constitutive model which was able to separate the thermal and athermal part of the flow stress. Since, in BCC materials, the effect of the viscous drag is noteworthy only when the strain rates are greater than 10^4^ s^−1^ [225], the authors did not consider the viscous drag effect, whereas they expressed the flow stress as a function of strain rate, temperature, strain, and microstructure and presented it mathematically as follows:(14)σ=σa*εn+1d(Kf+Vp(Kp−Kf))+σth^[1−{−KTGolnε˙εr˙}1q]1′p
where $\sigma_{a}^{*}$ is the material constant and is expressed as follows:(15)$$\sigma_{a}^{*}=\frac{\sigma_{a}-\sigma_{o}}{\varepsilon^{n}}$$
σ_o_ is mathematically expressed as follows:(16)$$\sigma_{o}=V_{f}\frac{K_{f}}{\sqrt{d_{f}}}+V_{p}\frac{K_{p}}{\sqrt{d_{p}}}$$
σth^ is the thermal stress at the reference temperature; *K* is the Boltzmann’s constant; *G*_0_ is the activation energy of 1ev/atom; $\varepsilon^{˙}$ is the strain rate; $\varepsilon_{r}^{˙}$ is the reference strain rate; *n* is the strain hardening exponent; *V_p_*, *K_p_*, and *d_p_* are the volume fraction, Hall–Petch constant, and grain size of the pearlite phase, respectively; *V_f_*, *K_f_*, and *d_f_* are the volume fraction, Hall–Petch constant, and grain size of the ferrite phase, respectively; *d* is the average grain size of ferrite and pearlite phases; and *p* and *q* represent the dislocation barrier profile [214].

During high strain-rate testing, a large amount of energy is converted into heat during plastic deformation of the material, leading to adiabatic conditions. Spirdione et al. [224] considered this fact and represented the results for both isothermal and adiabatic conditions by coupling the temperature effect in Equation (16). The temperature effect is expressed as follows:(17)$$\Delta T\approx \frac{\eta }{\rho }\int_{0}^{\gamma }\frac{\tau }{C_{v}}d\gamma$$
where the notations have their own meanings as mentioned earlier. The model parameters are shown in Table 4, and the comparison of the experimental values with the simulated results are shown in Figure 37. The figure represents a satisfying correlation between the experimental and simulation results.

Predicting the flow behaviour of the material during hot-working operations is often complex. Factors such as strain, strain rate, and temperature significantly affect the phenomenon of strain hardening and softening [205]. Noticeable research work has been done in this field for developing constitutive equations to predict the flow behaviour of the material during their hot-working deformation. Huang et al. [220] developed a model based on irreversible thermodynamics to predict the stress–strain behaviour of low and medium carbon steel during their hot-deformation condition as a function of strain rate and temperature and reported a satisfactory agreement between the experimental results and the predicted values. Serajzadeh et al. [226] developed and proposed a constitutive model which could predict the dynamic transformations and the flow stress behaviour as a function of strain rate and temperature. They verified their model by performing hot compression tests for two different grades of plain carbon steel of 0.08% and 0.7%C and confirmed the robustness and the validity of the proposed model. Yoshino et al. [227] investigated the effects of temperature, deformation, and heat treatment history on the flow stress behaviour of different types of carbon steels and developed a new equation based on the theory of generalized deformation energy. The flow stress equation is mathematically expressed as follows: (18)$$\sigma =\left\{{\left(\frac{\varepsilon^{˙}}{\varepsilon_{st}^{˙}}\right)}^{m}\sigma_{st}-B\right\}exp\left(-p{\left(\frac{\varepsilon^{˙}}{\varepsilon_{st}^{˙}}\right)}^{q}\left(T-T_{st}\right)\right)$$
where $\varepsilon^{˙}$ is the strain rate, $\varepsilon_{st}^{˙}$ is the reference state strain rate, $\sigma_{st}$ is the stress rate at the reference state, *m* is the strain rate sensitivity index, *p* and *q* are the material constants, and $T\;and\;T_{st}$ are absolute and reference temperature, respectively. They reported that the reference stress is dependent on the deformation energy, which on the other hand is the only parameter correlating the theory of plasticity with the structural changes in the lattice system. The developed model showed a good correlation between the predicted results and the experimental data. Marciniak and Konieczny [228], in their study, developed a model by taking into consideration the deformation history of a low carbon (<0.1% C) steel to predict its yield stress for a wide range of temperature and strain rates. They found the predicted model to be suitable for determining the yield stress of the material at high strain rates and temperature during the hot-working process. 

The simultaneous occurrence of DRV and DRX during hot deformation affects the flow stress behaviour of the material [104,229]. Mirzaie et al. [230] in their investigation employed the Zerilli–Armstrong (ZA) model to predict the flow stress behaviour of 0.50C-0.68Mn-0.20Si-0.28Cu steel at different strain rates and temperatures and observed the inability of the original model to predict the softening part of the flow stress. They modified the ZA model by including the peak strain and the effect of work-hardening and -softening phenomenon. The framed constitutive equation was claimed to be useful for modelling the hot deformation behaviour of plain carbon steel. Based on Equation (10) and after finding all the constants, Reference [159] expressed the ZA equation as follows: (19)$$\sigma =11697.5\varepsilon_{pl}^{0.5}\exp\left[\left(-0.003069+0.00012ln\varepsilon ˙\right)T\right]$$

The flow stress behaviour shown in Figure 38 is presented based on the predictions using Equation (21). The figure clearly shows that the model could not represent both the hardening and softening stages of the flow curve. They modified the existing ZA model by including the peak strain term and reported the requirement of two equations to effectively represent the strain-hardening and -softening parts of the curves. They mathematically represented their proposed model after calculating all the constants as follows:(20a)$$\sigma =9880.4{\left(\frac{\varepsilon }{\varepsilon_{pl}}\right)}^{n1}\exp\left[\left(-0.0034+0.00012ln\varepsilon ˙\right)T\right],\;\mathrm{if}\;\varepsilon \;\leq \;\varepsilon_{pl}$$
(20b)$$\sigma =9880.4{\left(\frac{\varepsilon }{\varepsilon_{pl}}\right)}^{n2}\exp\left[\left(-0.0034+0.00012ln\varepsilon ˙\right)T\right],\;\mathrm{if}\;\varepsilon \;\geq \;\varepsilon_{pl}$$
where *n*_1_ and *n*_2_ are the strain-hardening exponents for two different conditions and their values were obtained to be 0.32 and −0.173, respectively. The major drawback of their proposed model was that the predicted curves were not smooth due to the peak strain value. The calculated flow stress based on the proposed equation is shown in Figure 38b.

Testa et al. [231] developed a physical-based model to determine the yield stress as a function of temperature and strain rate for a low carbon A508 steel. Lee et al. [232] modified the Shida’s constitutive equation by changing the strain-rate hardening function to predict the flow stress behaviour of 0.1% carbon steel at elevated temperatures when the strain rates are higher beyond 100 s^−1^. They validated their proposed model by conducting mechanical testing in the SHPB apparatus at high temperatures ranging from 750 °C to 1050 °C and reported having a reasonable agreement between the experimental results and the predicted results. To predict the deformation behaviour of a hypo eutectoid pearlitic steel and low carbon (0.122% C) steel bimetal, Gao et al. [95] conducted strain-rate experiments on Gleeble 3500 simulator at wide range of temperature (800–1100 °C) and subsequently developed a constitutive model by incorporating the Zener–Hollomon parameter and other material constants to the Arrhenius-type equation. They expressed the flow stress as follows:(21)$$\sigma \left(T,\;\varepsilon ,\;\varepsilon^{˙}\right)=\frac{1}{\alpha \left(\varepsilon \right)}ln\left\{{\left(\frac{Z\left[T,\;\varepsilon^{˙},\;Q\left(\varepsilon \right)\right]}{A\left(\varepsilon \right)}\right)}^{\frac{1}{n\left(\varepsilon \right)}}+{\left[{\left(\frac{Z\left[T,\;\varepsilon^{˙},\;Q\left(\varepsilon \right)\right]}{A\left(\varepsilon \right)}\right)}^{\frac{2}{n\left(\varepsilon \right)}}+1\right]}^{1/2}\right\}$$
where *α* is the material parameter and is known as the stress multiplier, *A* is the material constant, n is the stress exponent, and the other notations have their usual meanings as mentioned earlier. The developed model showed a good agreement with the experimental results, as shown in Figure 39.

The effects of strain rate on the deformation behaviour of DP steel have been previously studied [131,233,234,235,236,237]. Roth et al. [233] coupled a JC model with the Swift–Voce strain-hardening model to predict the strain-rate deformation behaviour of DP800 steel and found a reasonable correlation between the experimental results and the predicted values. The proposed model was capable of considering the effect of thermal softening as the temperature was considered to be an internal variable. Gruben et al. [131] characterise the flow behaviour of a Docol 600DL dual-phase steel combining the isotropic hardening and Hershey yield function. The authors conducted uniaxial tests to determine the material parameters and reported a good correlation between the experimental and simulation results. Sarraf et al. [234] used the Rousselier ductile damage model to characterise the strain hardening and plastic instability of DP 600 steel for strain rates ranging from 10^−3^ to 10^3^ s^−1^. Song et al. [235] used the existing JC equation and optimized the existing ZA model by introducing a strain-rate sensitivity parameter (m) with five material constants (*σ_o_*, *c*_1_, *c*_5_, *m*, and *n)* to describe the flow behaviour of DP1000 steel ranging from 10^−4^ to 2000 s^−1^. The authors further reported a better correlation and fitting of the adapted model compared to the Johnson–Cook model. Furthermore, an increase in the yield strength and ultimate tensile strength of the material was observed with an increase in the strain rate. The optimized model was mathematically expressed as follows:(22)$$\sigma =\sigma_{0}+c_{1}\varepsilon_{pl}^{m}+c_{5}\varepsilon_{pl}^{n}$$

Cao et al. [236] characterise the tensile flow behaviour of DP800 steel by using the Voce equation. The authors separated the thermal and athermal part of the flow stress and reported a significant contribution of the thermal stress during the strain rate deformation behaviour of DP800 at elevated temperatures. Qin et al. [40] predicted the stress–strain behaviour of DP700 and DP500 steel by using the JC model and found the experimental results to be in good agreement with the predicted numerical values. Tarigopula et al. [162] developed a constitutive model to represent the strain rate behaviour of DP800 steel. The authors described the effective stress (*σ*) by modifying the Voce rule as follows:(23)$$\sigma =\sigma_{0}+\overset{2}{\underset{i=1}{\sum }}Q_{i}[1-\exp(-C_{i}\varepsilon^{p})]\left[{\left(1+\frac{\varepsilon^{˙p}}{\varepsilon_{o}{}^{˙}}\right)}^{q}\right]$$
where the parameters *σ*_0_, *Q_i_*, *C_i_*, and *q* are the material constants; *ε^p^* is the effective plastic strain; *ε_o_*^.^ is the reference strain rate defined by the user; and *ε^.p^* is the effective plastic strain rate. The simulation and experimental results showed a good correlation between each other in terms of strain hardening and strain-rate hardening, as shown in Figure 40.

## 4. Concluding Remarks

This review article has focused on the strain-rate deformation behaviour of plain carbon steel during metal forming and high impact loading conditions. The article emphasizes three important aspects which include (a) mechanical tests from quasi-static to high strain rates, (b) structure–property correlation of plain carbon steels, and (c) strength and limitations of phenomenological and physical-based constitutive equations. An effort has been made to explain the significance and impact of the structural evolution of plain carbon steel on its mechanical properties. Published research works on the high strain-rate deformation behaviour of high carbon steel are very scarce. This may be due to their relatively brittle nature and the lack of interest in understanding their deformation mechanism at high strain rates. However, there are particular cases of application of high carbon steel such as in ball mills where the grinding balls are subjected to varying strain rates from 10^−3^ to 10^4^ s^−1^ and thus require a fundamental understanding of their deformation behaviour.

## Figures and Tables

**Figure 1 materials-13-00556-f001:**
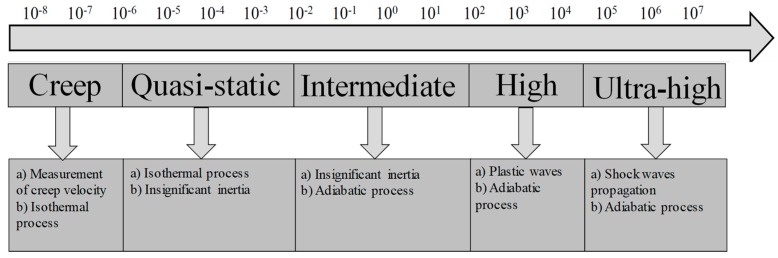
Regimes of strain rate.

**Figure 2 materials-13-00556-f002:**
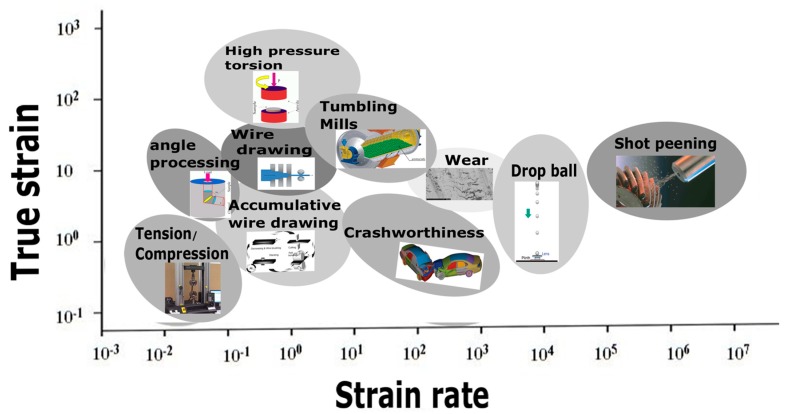
Range of strain rates and strain accumulation in various deformation processes.

**Figure 3 materials-13-00556-f003:**
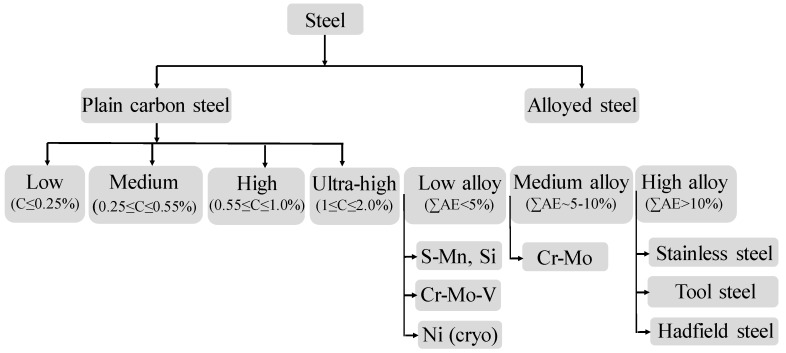
Schematic chart for the classification of steel.

**Figure 4 materials-13-00556-f004:**
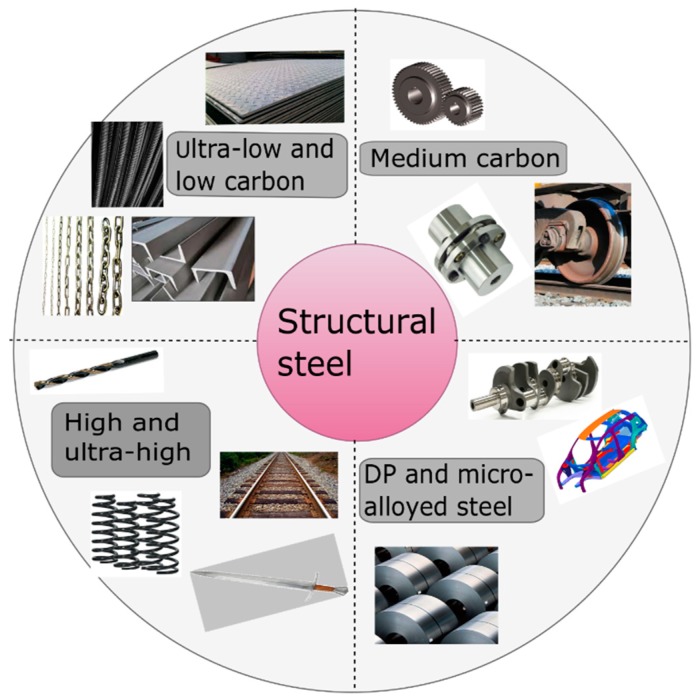
Applications of structural steels.

**Figure 5 materials-13-00556-f005:**
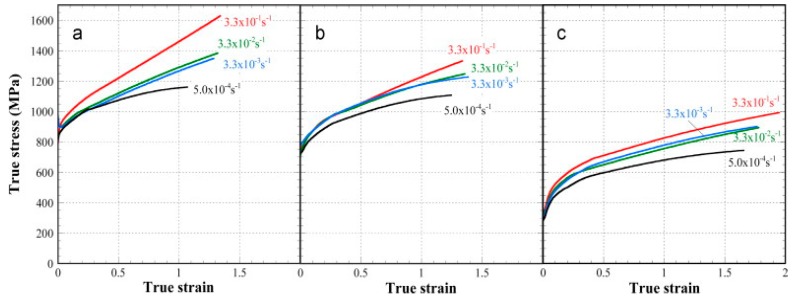
True stress–strain curve of the ferrite-cementite (FC) steels with (**a**) ferrite grain size of 0.5 μm, (**b**) ferrite grain size of 0.8 μm, and (**c**) ferrite grain size of 34 μm at different strain rates [58].

**Figure 6 materials-13-00556-f006:**
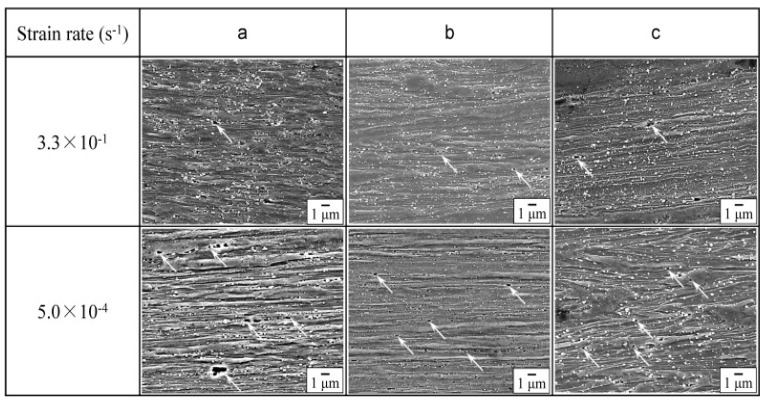
SEM images of the cross-sectional planes of the FC steel beneath the fracture surface at different strain rates of 3.3 × 10^−1^/s and 5.0 × 10^−4^/s respectively [58].

**Figure 7 materials-13-00556-f007:**
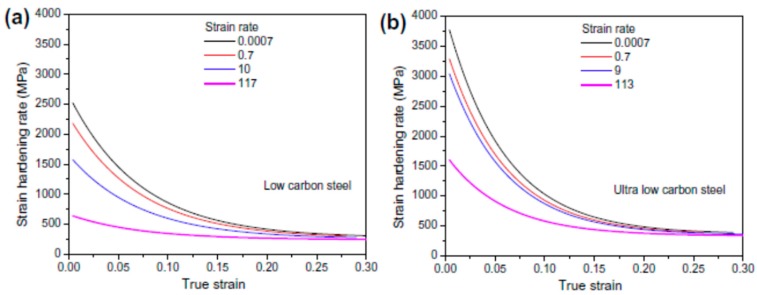
Strain-hardening rate at various strain rates for (**a**) low carbon steel and (**b**) ultralow carbon steel [22].

**Figure 8 materials-13-00556-f008:**
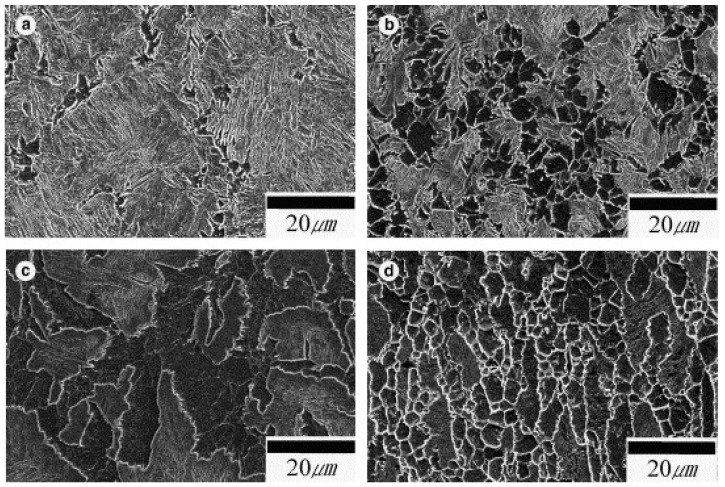
SEM micrographs showing the evolution of ferrite grain structure at (**a**) 800 °C and ε = 0.5; (**b**) 800 °C and ε = 0.8; (**c**) 750 °C and ε = 0.25; and (**d**) 750 °C and ε = 0.7 [66].

**Figure 9 materials-13-00556-f009:**
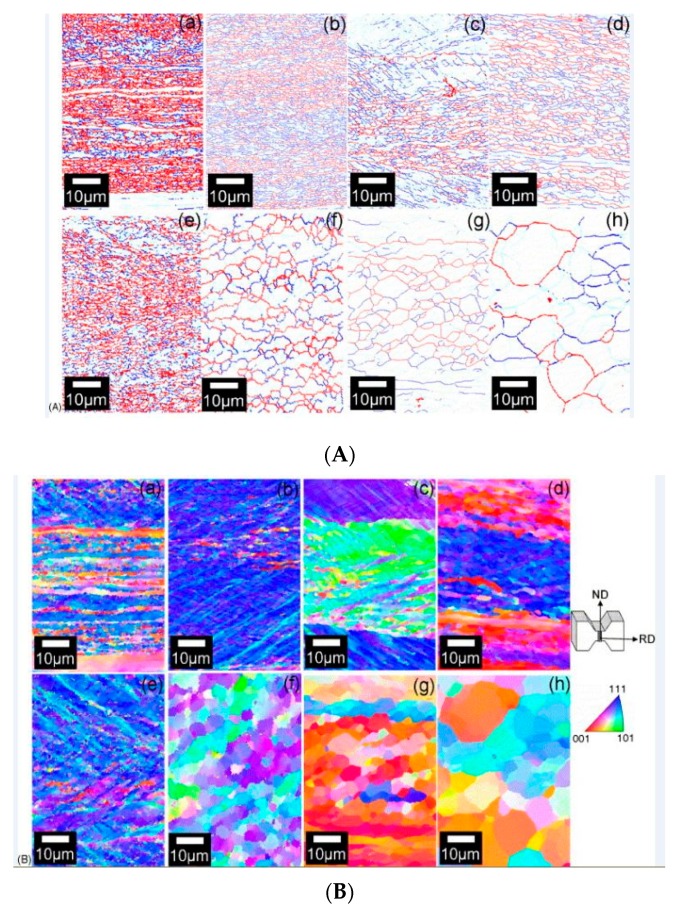
(**A**) Boundary maps of the specimens deformed at a strain rate of 1 s^−1^ at (**a**) 823 K, (**b**) 873 K, (**c**) 923 K, (**d**) 973 K, (**e**) 773 K, (**f**) 873 K, (**g**) 923 K, and (**h**) 1023 K observed at a strain of 4 [77]. (**B**) Crystallographic orientation distribution of local regions along the normal direction (ND) obtained by electron backscatter diffraction (EBSD) for specimens deformed at a strain rate of 1 s^−1^ (top) at (**a**) 823 K, (**b**) 873 K, (**c**) 923 K, (**d**) 973 K, (**e**) 773 K, (**f**) 873 K, (**g**) 923 K, and (**h**) 1023 K observed at a strain of 4 [77].

**Figure 10 materials-13-00556-f010:**
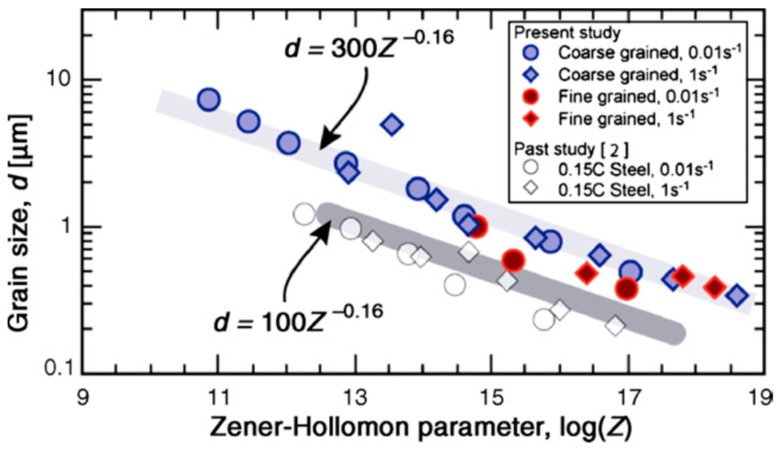
Variation of the grain size with Z parameter at a strain of 4 obtained from boundary maps: The grain size data of 0.15 C steel is also plotted for comparison [77].

**Figure 11 materials-13-00556-f011:**
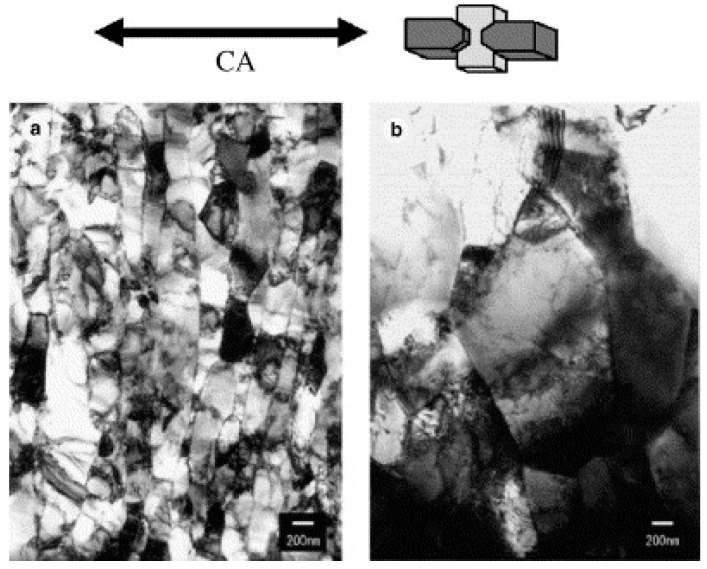
TEM images of the specimen deformed at 823 K at (**a**) 1 s^−1^ and (**b**) 0.01 s^−1^ [78].

**Figure 12 materials-13-00556-f012:**
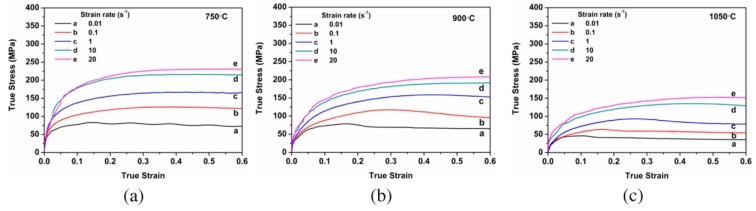
Flow curves of AISI 1010 steel in compression obtained using different strain rates after austenitization at 1050 °C for 5 min and at deformation temperatures of (**a**) 750 °C, (**b**) 900 °C, and (**c**) 1050 °C [80].

**Figure 13 materials-13-00556-f013:**
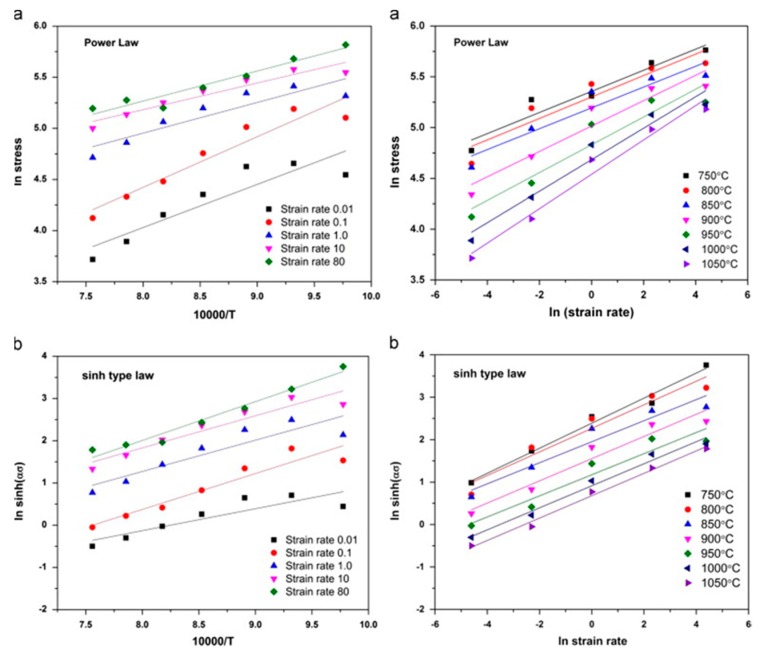
Flow stress and temperature plots for all strain rates at the true strain of 0.6 using (**a**) power law and (**b**) sinh type law [82].

**Figure 14 materials-13-00556-f014:**
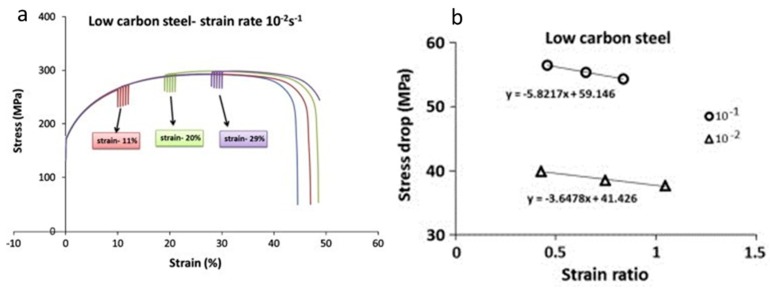
(**a**) Comparison of stress relaxation and monotonic tensile curve; (**b**) stress drop versus strain ratio [99].

**Figure 15 materials-13-00556-f015:**
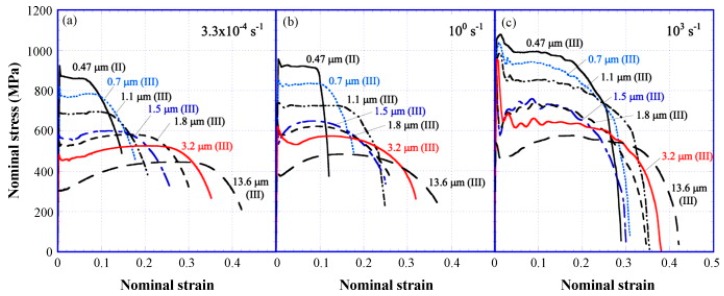
Nominal stress–strain curves of FC specimens obtained by tensile tests with strain rates of 3.3 × 10^−4^ s^−1^ (**a**), 100 s^−1^ (**b**), and 10^3^ s^−1^ (**c**) at 296 K [100].

**Figure 16 materials-13-00556-f016:**
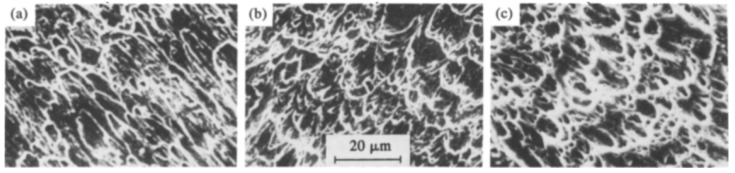
Fractographs of different steels: (**a**) quenched steel, (**b**) quenched and tempered steel, and (**c**) normalized steels [102].

**Figure 17 materials-13-00556-f017:**
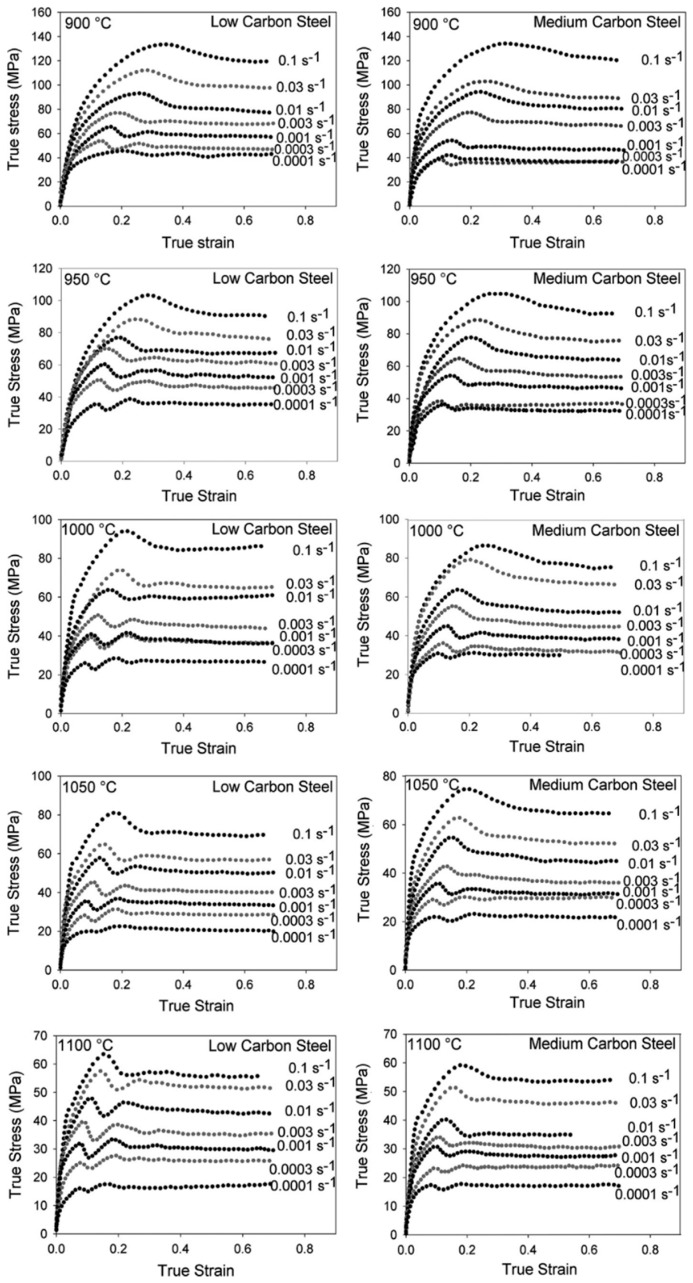
Flow curves at different deformation conditions [103].

**Figure 18 materials-13-00556-f018:**
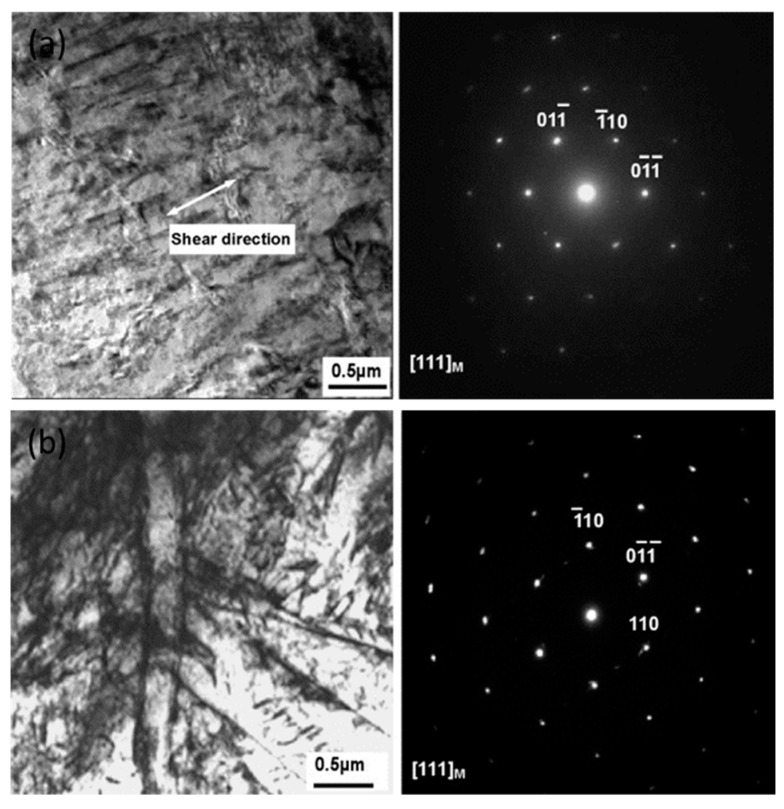
TEM and selected area diffraction (SAD) pattern of (**a**) deformed shear bands and (**b**) transformed shear bands [107].

**Figure 19 materials-13-00556-f019:**
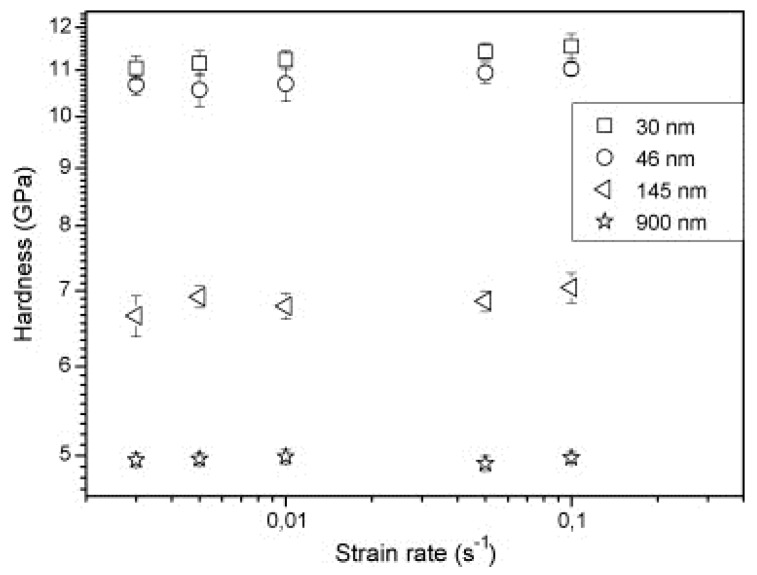
Nanoindentation hardness vs. strain rate for the different grain sizes [114].

**Figure 20 materials-13-00556-f020:**
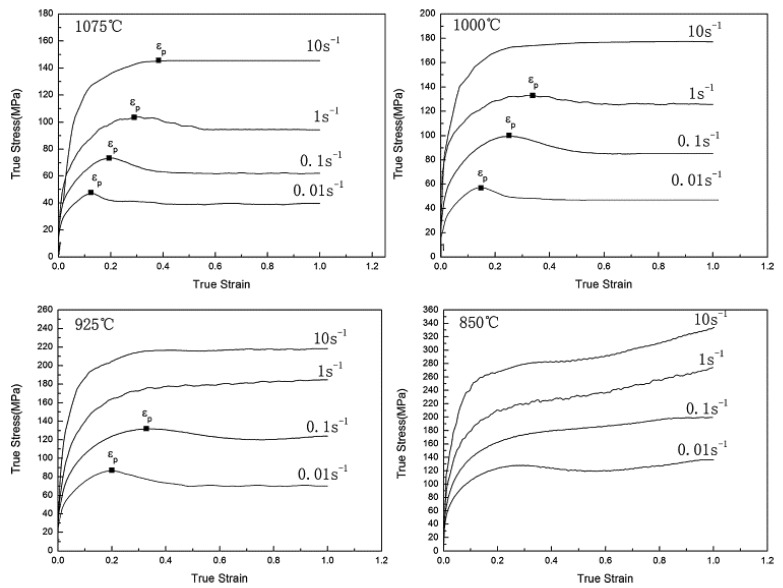
True stress–strain curve under different temperature and strain rates [115].

**Figure 21 materials-13-00556-f021:**
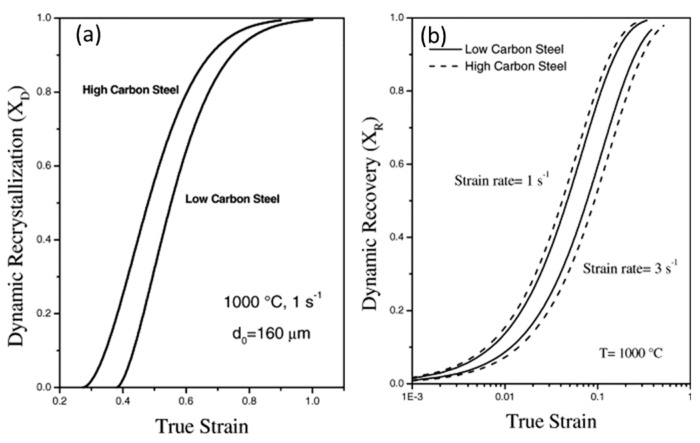
The progress of (**a**) dynamic recrystallization and (**b**) dynamic recovery at 1000 °C [117].

**Figure 22 materials-13-00556-f022:**
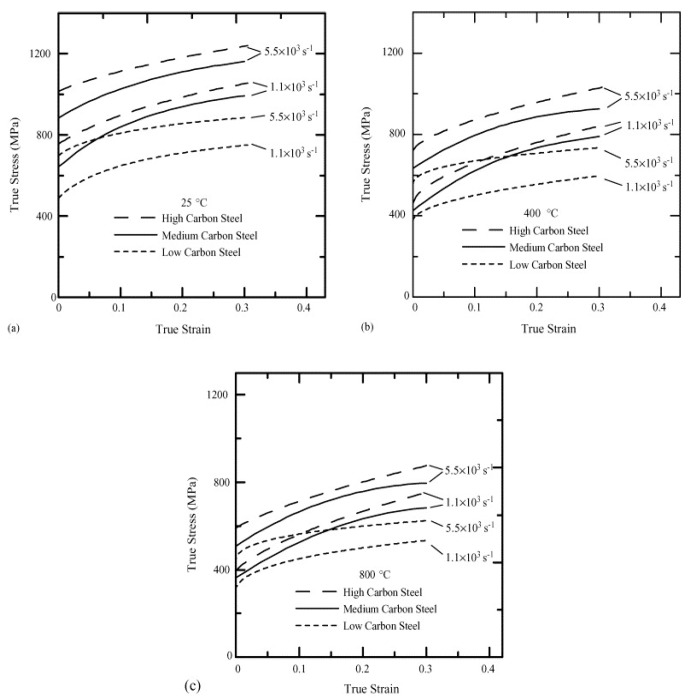
Stress–strain curves of S15C, S50C, and SKS93 at temperatures of (**a**) 25 °C; (**b**) 400 °C; and (**c**) 800 °C [123].

**Figure 23 materials-13-00556-f023:**
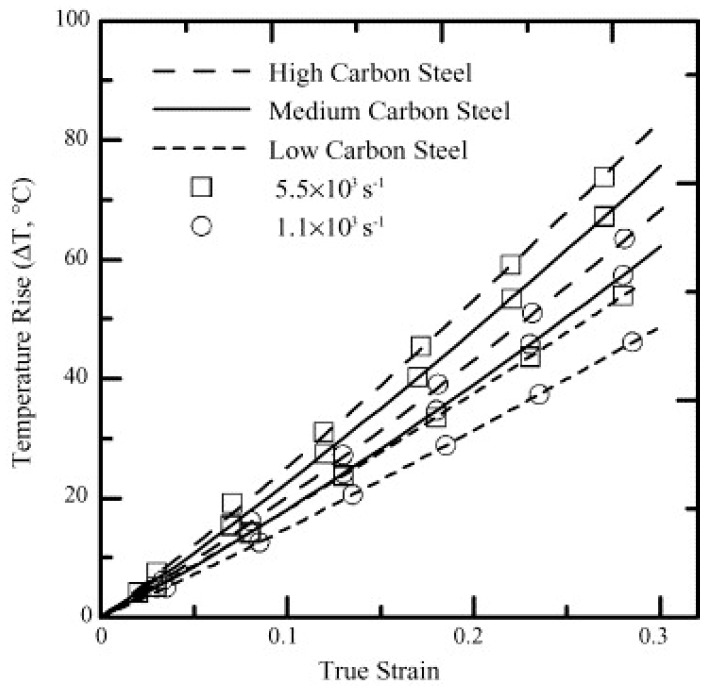
Temperature vs. true strain for three kinds of steels at different strain rates [123].

**Figure 24 materials-13-00556-f024:**
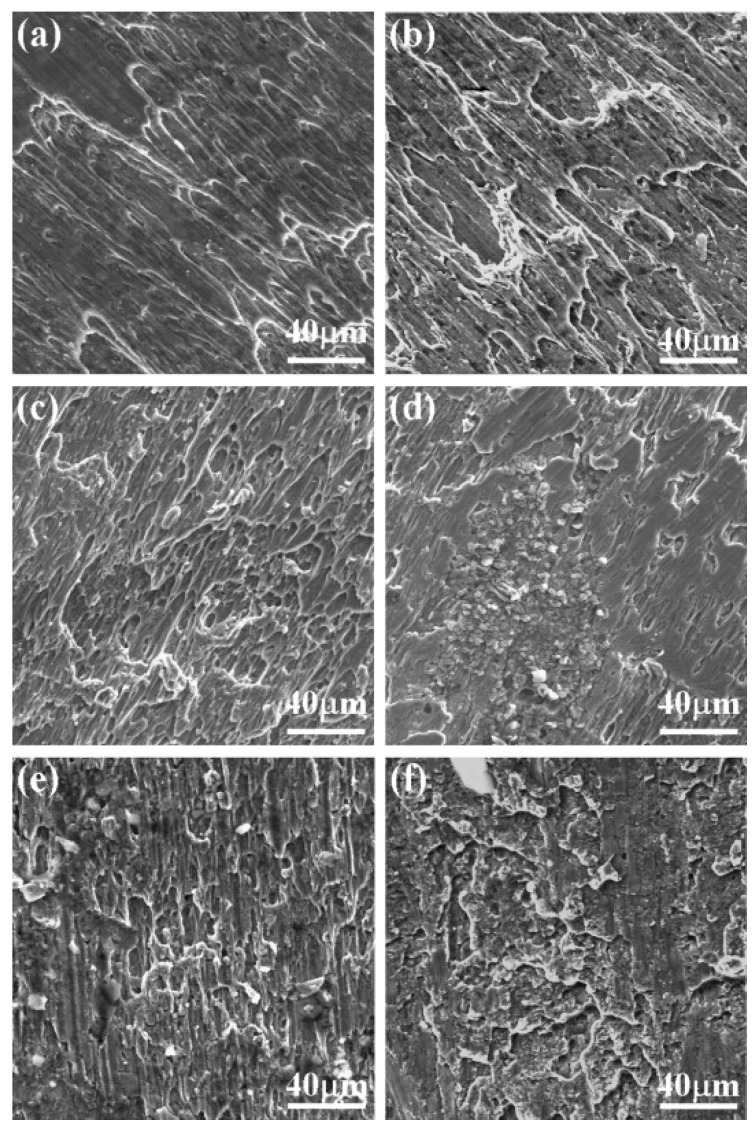
Fracture surfaces of S15C steel deformed at (**a**) 5 × 10^4^ s^−1^ and (**b**) 2 × 10^5^ s^−1^; S50C steel deformed at (**c**) 5 × 10^4^ s^−1^ and (**d**) 2 × 10^5^ s^−1^; and SKS93 steel deformed at (**e**) 5 × 10^4^ s^−1^ and (**f**) 2 × 10^5^ s^−1^ [126].

**Figure 25 materials-13-00556-f025:**
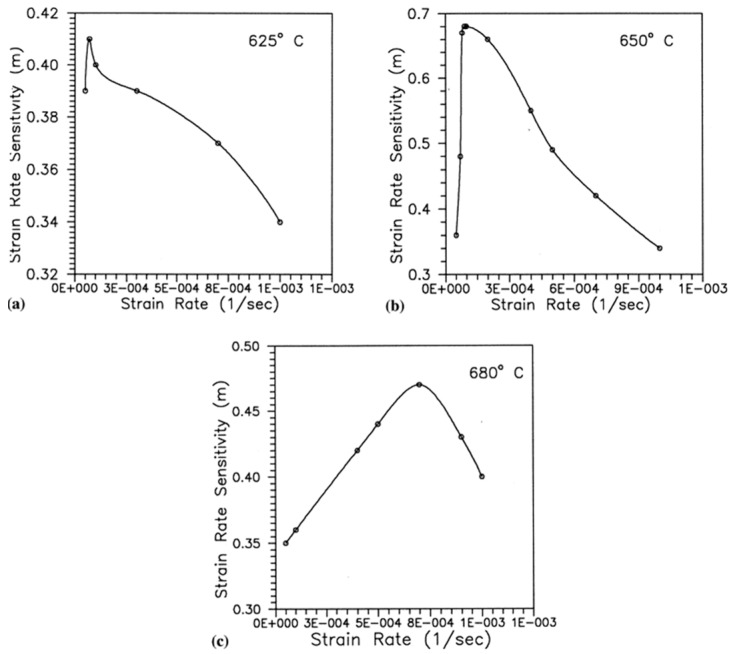
The effect of strain rate on strain-rate sensitivity: (**a**) 625 °C, (**b**) 650 °C, and (**c**) 680 °C [134].

**Figure 26 materials-13-00556-f026:**
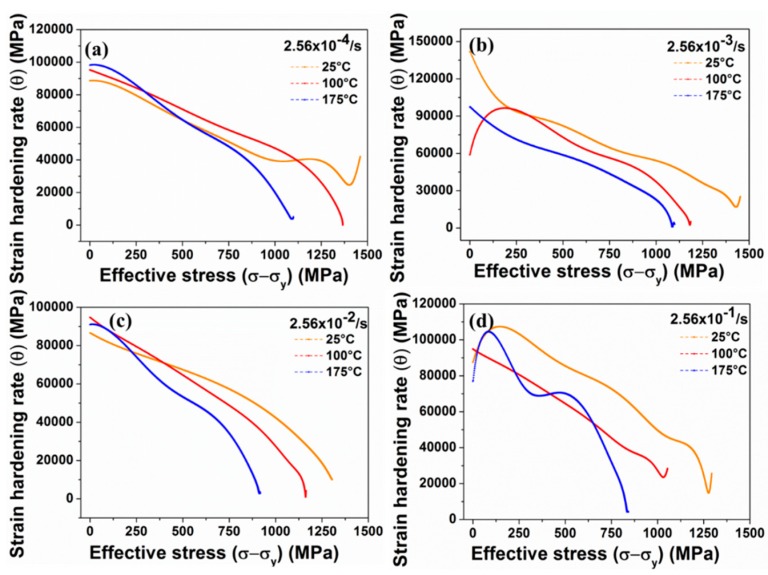
Strain-hardening rate as a function of effective stress at different temperatures and strain rates of (**a**) 2.56 × 10^−4^/s, (**b**) 2.56 × 10^−3^/s, (**c**) 2.56 × 10^−2^/s, and (**d**) 2.56 × 10^−1^/s [140].

**Figure 27 materials-13-00556-f027:**
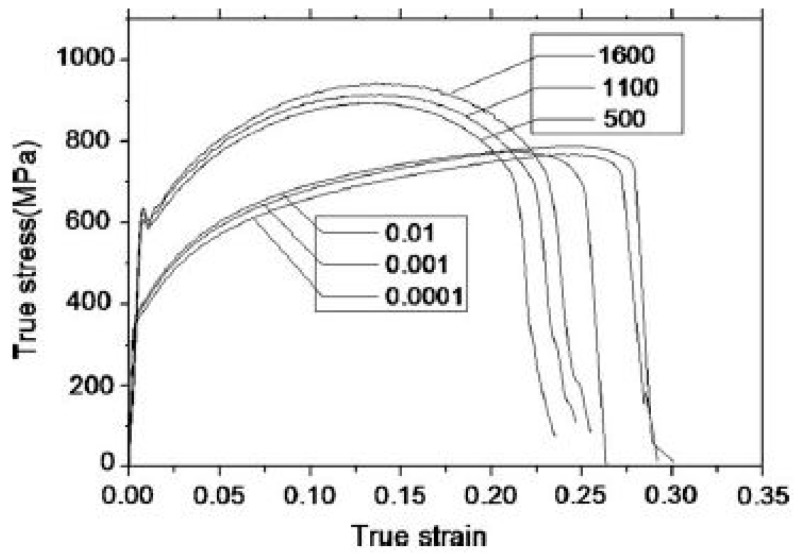
True stress–strain curve at different strain rates [159].

**Figure 28 materials-13-00556-f028:**
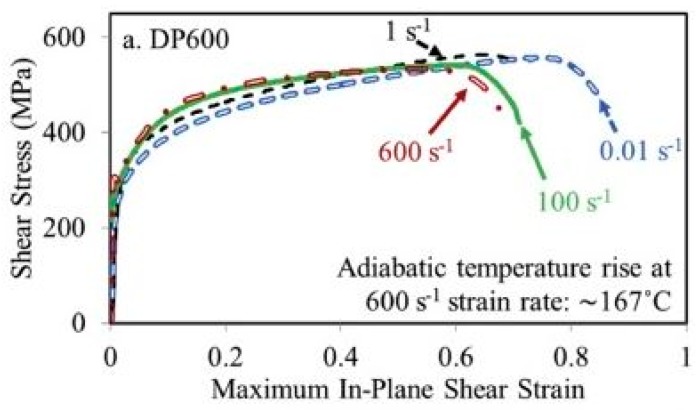
Effect of strain rate on the deformation behaviour during shear experiments for DP600 steel [166].

**Figure 29 materials-13-00556-f029:**
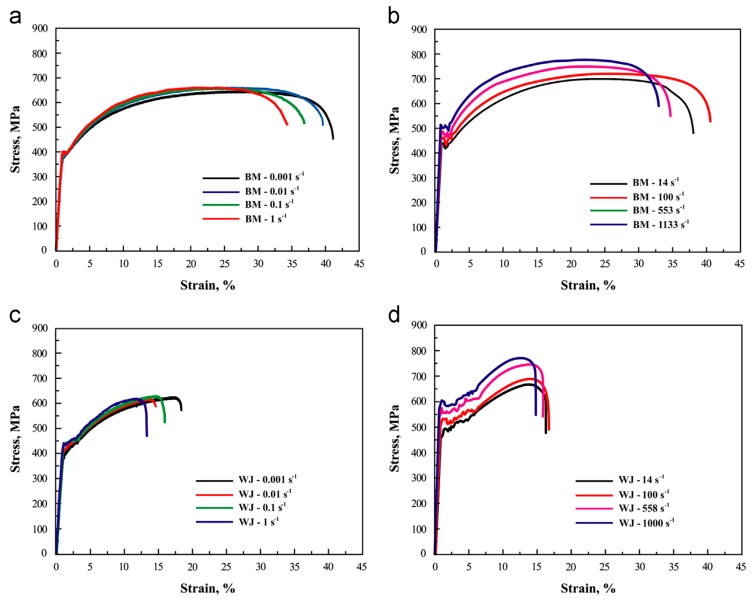
Stress–strain curves for (**a**) DP600 base metal (BM) tested at strain rates from 0.001 to 1 s^−1^, (**b**) DP600 BM tested at strain rates from 14 to 1133 s^−1^, (**c**) DP600WJ tested at strain rates from 0.001 to 1 s^−1^, and (**d**) DP600 WJ tested at strain rates from 14 to 1000 s^−1^ [171].

**Figure 30 materials-13-00556-f030:**
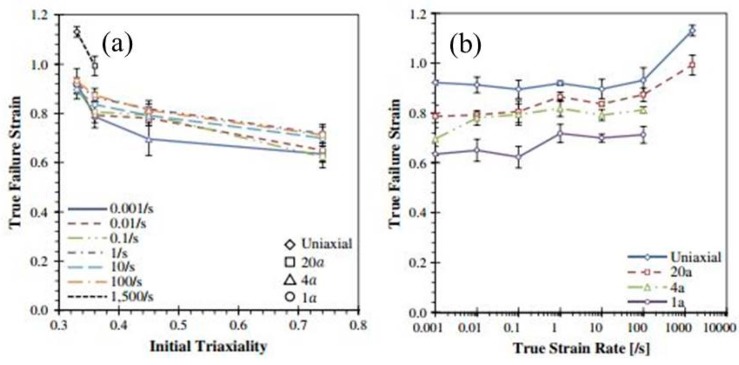
True failure strain as a function of (**a**) stress triaxiality and (**b**) true strain rate [175].

**Figure 31 materials-13-00556-f031:**
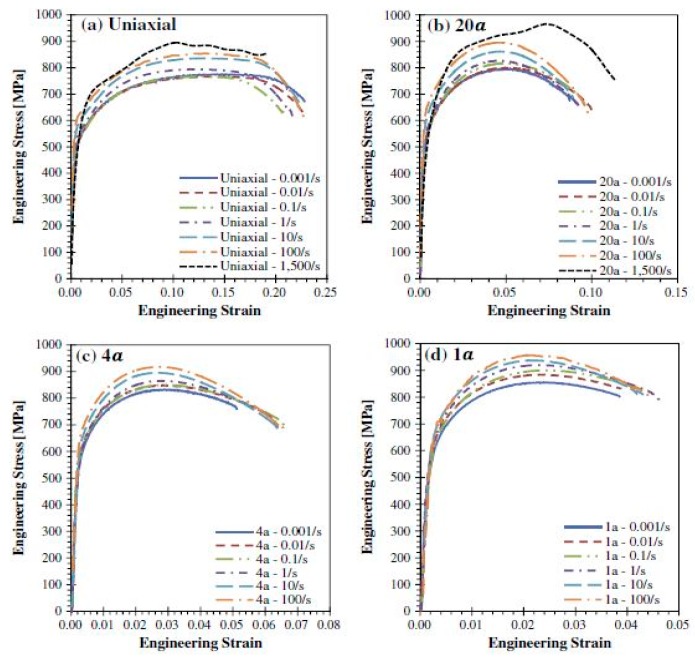
Stress strain curve of DP780 steel for (**a**) uniaxial test, (**b**) 17.5-mm notch specimen, (**c**) 3.5-mm notch specimen, and (**d**) 1-mm notch specimen [175].

**Figure 32 materials-13-00556-f032:**
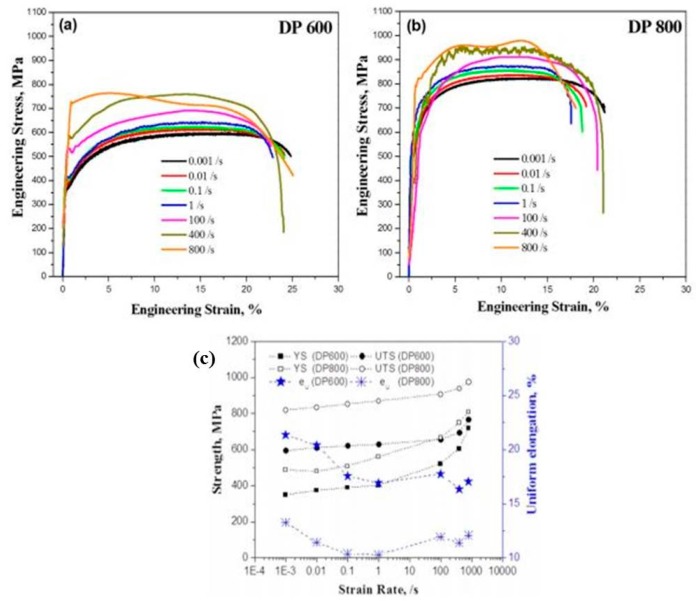
(**a**) Stress–strain curves for dual-phase (DP) 600, (**b**) stress–strain curves for DP 800, and (**c**) variation in the yield and ultimate tensile strength of DP600 and DP800 steel as a function of strain rate [178].

**Figure 33 materials-13-00556-f033:**
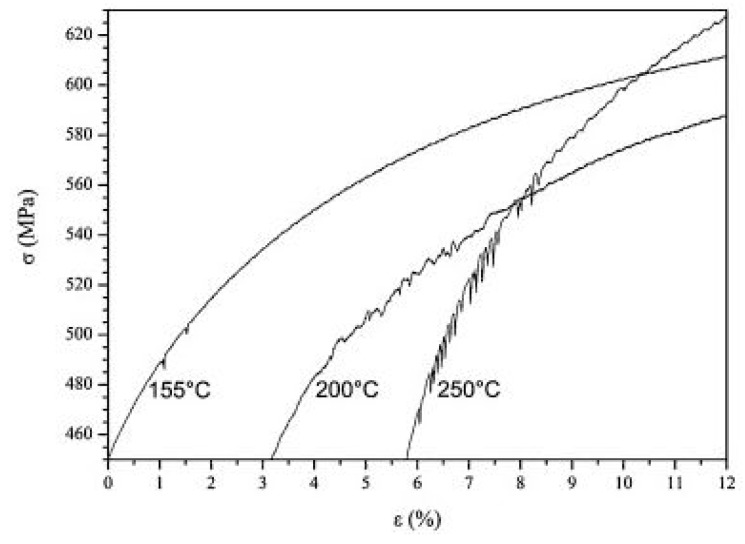
Serrations in the stress–strain curves at a strain rate of 10^−3^ s^−1^ at different temperature for 0.10%C DP steel [194].

**Figure 34 materials-13-00556-f034:**
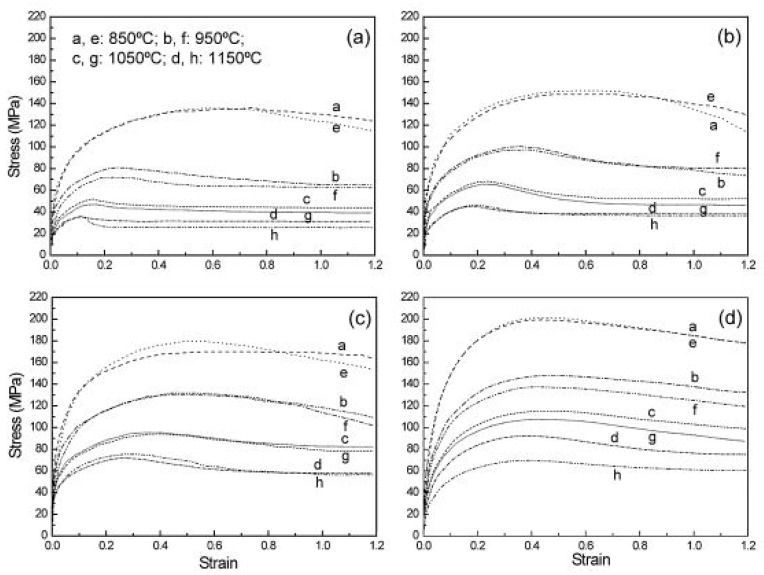
Stress–strain curves of air-cooled (a–d) and water-cooled (e–h) Al–V–N micro-alloyed steel after solution treatment deformed at various strain rates: (**a**) 0.01 s^−1^; (**b**) 0.1 s^−1^; (**c**) 1 s^−1^; and (**d**) 5 s^−1^ [200].

**Figure 35 materials-13-00556-f035:**
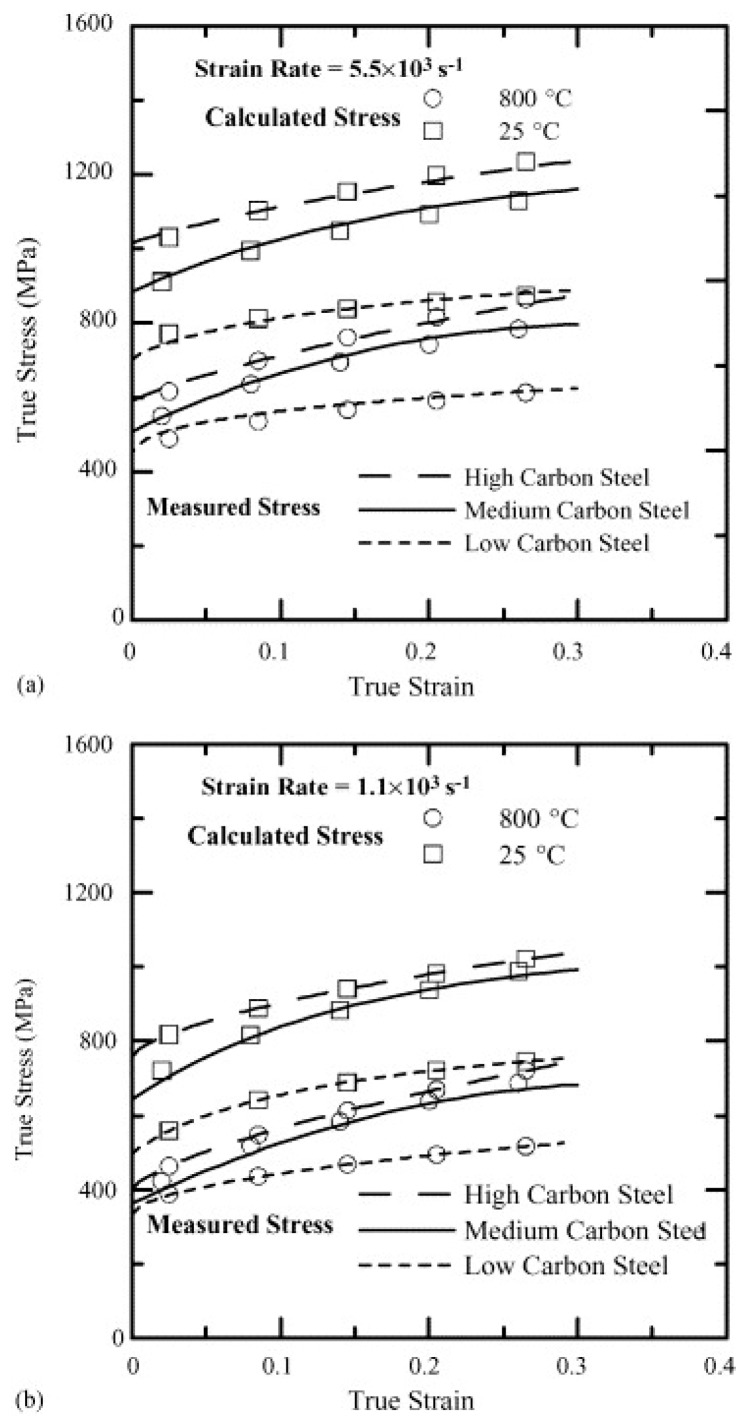
Comparison of calculated and experimental flow curves at two different conditions at (**a**) 5.5 × 10^−3^ s^−1^ and (**b**) 1.1 × 10^−3^ s^−1^ [123].

**Figure 36 materials-13-00556-f036:**
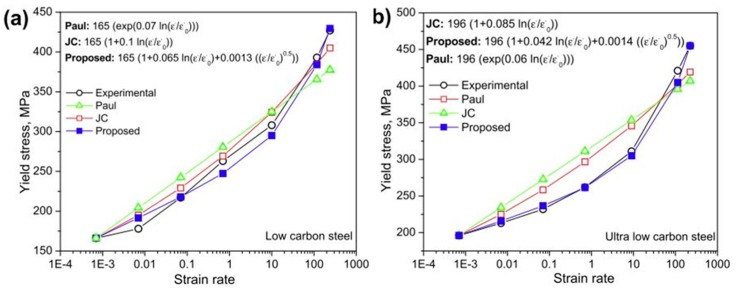
Comparison of the predicted and experimental yield strength at varying strain rates for (**a**) low carbon steel and (**b**) ultralow carbon steel [22].

**Figure 37 materials-13-00556-f037:**
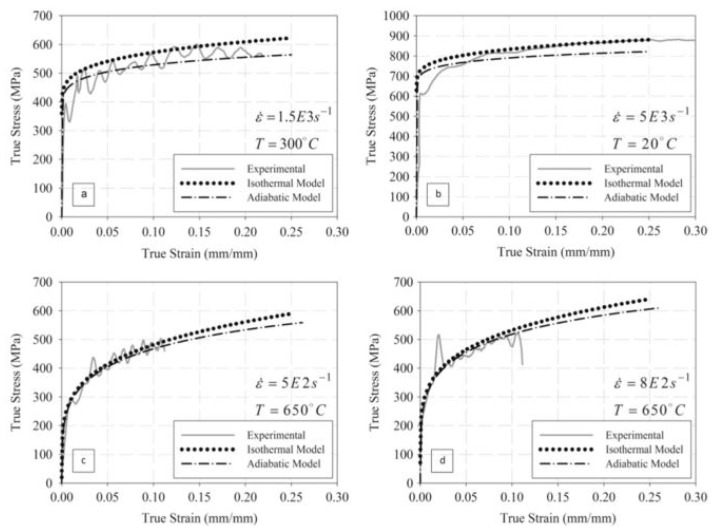
Comparison of experimental and simulation results for adiabatic and isobaric conditions for as-received material at (**a**) 1.5 × 10^3^ s^−1^ at 300 °C and (**b**) 5 × 10^3^ s^−1^ at 300 °C and for the heat treated material at (**c**) 5 × 10^2^ s^−1^ at 6500 °C and (**d**) 8 × 10^2^ s^−1^ at 650 °C [224].

**Figure 38 materials-13-00556-f038:**
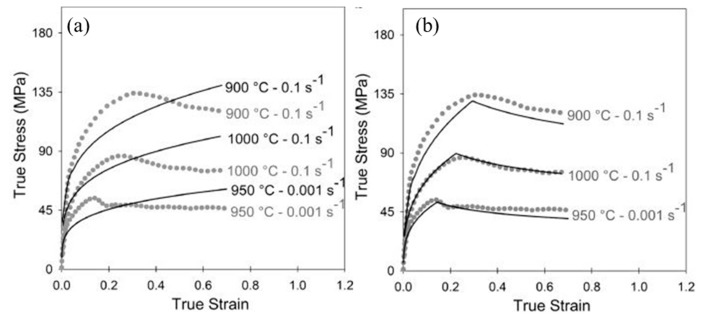
Comparison of the experimental and calculated values of true stress–strain based on (**a**) the existing Zerilli–Armstrong (ZA) model and (**b**) the modified ZA model [230].

**Figure 39 materials-13-00556-f039:**
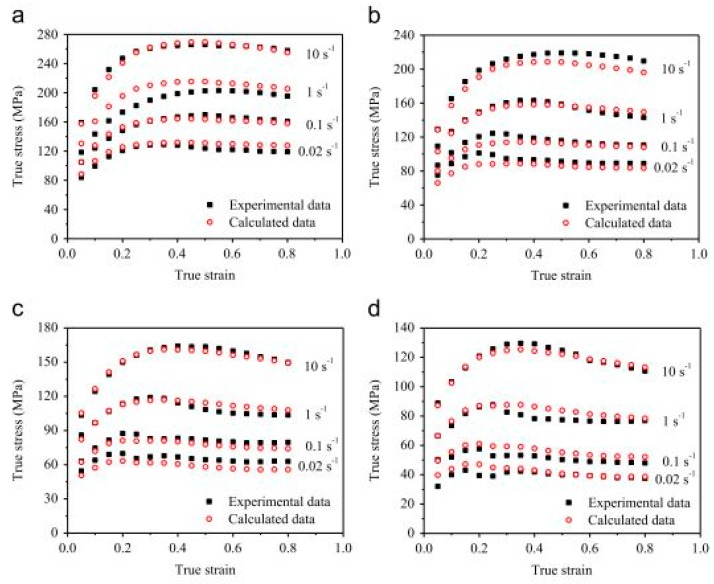
Comparison of the experimental and calculated stress–strain curves at different conditions [95].

**Figure 40 materials-13-00556-f040:**
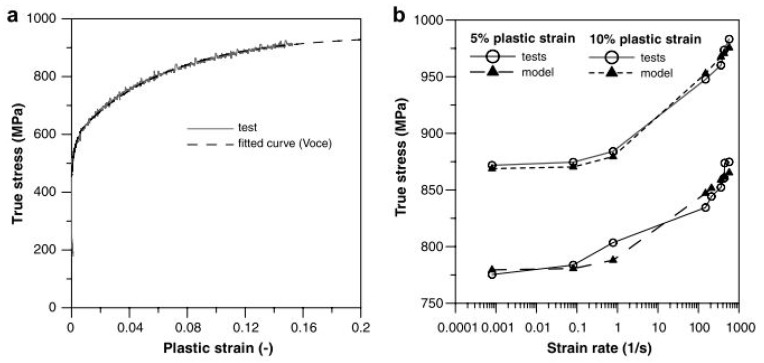
Comparison of calculated and experimental curves for DP800 steel for (**a**) strain hardening and (**b**) strain-rate hardening [162].

**Table 1 materials-13-00556-t001:** Test methods for different regimes of strain rate [14].

Strain Rate (s^−1^)	Experimental Techniques
Compression Tests	
<0.1	Conventional Load Frames, Gleeble
0.1–100	Special Servo-hydraulic Frames, Gleeble
0.1–500	Cam Plastometer and Drop Test
200–10^4^	Split Hopkinson Pressure Bar (SHPB)
10^3^–10^5^	Taylor Impact Test
>10^5^	Single and two-stage gas gun
Tension Tests	
<0.1	Conventional Load Frames, Gleeble
0.1–100	Special Servo hydraulic Frames, Gleeble
100–10^3^	Split Hopkinson Pressure Bar (in tension)
10^4^	Expanding Ring
>10^5^	Flyer Plate
Shear and Multiaxial Tests	
<0.1	Conventional Shear Tests
0.1–100	Special Servo-hydraulic Frames
10–10^3^	Torsional Impact
100–10^4^	Split Hopkinson Pressure Bar (in torsion)
10^3^–10^4^	Double-notch Shear and Punch
10^4^–10^7^	Pressure-shear Plate Impact

**Table 2 materials-13-00556-t002:** Calculated values of Zerilli–Armstrong coefficients [123].

Carbon Steel	Coefficients					
	*c*_1_ (MPa)	*c*_2_ (MPa)	*c*_3_ (k^−1^)	*c*_5_ (k^−1^)	*c*_5_ (MPa)	*n*
S15C	315.75	615.38	0.013	0.0012	365	0.50
S50C	321.81	849.81	00997	0.000996	759.60	0.59
SK93	460.04	700.11	0.00142	0.00152	712.02	0.69

**Table 3 materials-13-00556-t003:** Material constant for the proposed model [22].

Steel	*σ* _o_	*B*	*C*	*β*	*A*	*K*	*G*	*H*
Ultralow carbon	190	360	215	17	0.042	0.0014	0.02	0.00045
Low carbon	165	270	170	14	0.065	0.0013	0.01	0.0009

**Table 4 materials-13-00556-t004:** Optimized values of the model parameters [224].

$\sigma_{a}^{*}$ (MPa)	*n*	*K*_p_ (MPa/μm^1/2^)	*K*_f_ (MPa/μm^1/2^)	σth^ (MPa)	*p*	*q*	*K* (eV/K)	*G*_0_ (eV)	$\varepsilon_{r}^{˙}$ (S^−1^)
346.5	0.2304	718.67	465	950	1	1.5	8.63 × 10^−5^	0.92	1.86 × 10^−8^

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
