# Peer review of "Structure–Property Correlation and Constitutive Description of Structural Steels during Hot Working and Strain Rate Deformation"

_materials, 2020, doi:10.3390/ma13030556_

Round 1

Reviewer 1 Report

Review for Materials- 670614

Structure-property correlation and constitutive description of plain carbon steels during hot working and strain rate deformation

The authors present an interesting and well-organized paper. Anyway, I present several recommendations:

In Figure 5 is reflected several applications of plain carbon and low alloy steels. As far as I know, railway rails are made of pearlitic steels, as you can see in several references, e.g. org/10.3390/met6050114; https://doi.org/10.1016/j.pmatsci.2016.06.001 Please, improve the visibility of the original figures (Figures 1, 2, 3, etc.). Please, write the reference section according to the MDPI format. Although the number and the selection of references is adequate, it would be advisable to include some papers from the journals of MDPI editorial (Metals, Coatings, Applied Sciences, Materials, etc.) related to the topic of the manuscript.

Reviewer 2 Report

The review is relatively complex, but the title and the terminology are a bit misleading. The review is in fact not only about the plain carbon steel, but about the structural (construction) steel in general. For example, the statement "Medium carbon steels are mostly used for the structural parts of machines, automotive industries and nuclear power plants." is not true, because the alloyed (or microalloyed) steel is used in most of the mentioned sectors. Similarly, the DP and CP advanced high-strength steels are also not considered as plain carbon steel. 

On the other hand, in Introduction, there are many misleading points in this paragraph: "All the metallic materials exhibit crystalline structure which forms during the solidification of the molten state of the material. Pure iron occurs in ferrite form at room temperature which is also known as α ferrite. At higher temperatures, the ferritic structure is unstable and gets converted into γ austenite. At even higher temperatures, the austenite might again transform into a higher temperature form of ferrite; this is called δ ferrite [48]. Carbon steel exists in three crystalline structures namely BCC (ferrite), FCC (austenite) and BCT (martensite)."

Not all metallic materials are crystalline. Do you know the so-called metallic glass?

Ferrite is not a form of the pure iron, but a solid solution of carbon in alpha-iron. Similarly, austenite is not a pure gamma-iron.

BCT (martensite) structure is not a stable form. It has to be stated.

It implies that the paper has to be rewritten completely. 

Round 2

Reviewer 2 Report

In the previous round, the problem of the paper was an incorrect terminology. It was corrected sufficiently.